# Biogenic and anthropogenic sources of aerosols at the high Arctic site Villum Research Station

Ingeborg E. Nielsen[1,2], Henrik Skov[1,2,5], Andreas Massling[1,2], Axel C. Eriksson[3,4],

Manuel Dall'Osto[6], Heikki Junninen[7,10], Nina Sarnela[7], Robert Lange[1,2], Sonya

Collier[8], Qi Zhang[8], Christopher D. Cappa[9] and Jacob K. Nøjgaard[1,2]*

[1]Department of Environmental Science, Aarhus University, Roskilde, 4000, Roskilde, Denmark
[2]Arctic Research Centre, Aarhus University, Aarhus, 8000, Aarhus, Denmark
[3]Division of Ergonomics and Aerosol Technology, Lund University, Box 118, SE-22100, Lund, Sweden
[4]Division of Nuclear Physics, Lund University, Lund, Box 118, SE-22100, Lund, Sweden
[5]Institute of Chemical Engineering and Biotechnology and Environmental Technology, University of Southern Denmark, 5230, Odense, Denmark
[6]Institute of Marine Sciences, CSIC, Passeig Marítim de la Barceloneta, 37-49. E-08003, Barcelona, Spain
[7]Institute for Atmospheric and Earth System Research / Physics, Faculty of Science, University of Helsinki, 00140 Helsinki, Finland
[8]Department of Environmental Toxicology, University of California, Davis, CA 95616, USA
[9]Department of Civil and Environmental Engineering, University of California, Davis, CA 95616, USA
[10]Institute of Physics, University of Tartu, Ülikooli 18, EE-50090 Tartu, Estonia

*Correspondence to:* Ingeborg Elbæk Nielsen (ien@envs.au.dk)

**Abstract**. There are limited measurements of the chemical composition, abundance, and sources of atmospheric particles in the high Arctic. To address this, we report 93 days of Soot Particle Aerosol Mass Spectrometer (SP-AMS) data collected from February 20th until May 23rd 2015 at Villum Research Station (VRS) in Northern Greenland (81°36' N).2 During this period, we observed the Arctic haze phenomenon with elevated $PM_1$ concentration ranging from an average of 2.3, 2.3 and 3.3 µg m$^{-3}$ in February, March and April to 1.2 µg m$^{-3}$ in May. Particulate sulfate ($SO_4^{2-}$) accounted for 66% of the non-refractory $PM_1$ with highest concentration until the end of April and decreasing in May. The second most abundant species was organic aerosol (OA) (24%). Both OA and $PM_1$, estimated from the sum of all collected species, showed a marked decrease throughout May in accordance with the polar front moving North together with changes in aerosol removal processes. The highest refractory black carbon (rBC) concentrations were found in the first month of the campaign averaging 0.2 µg m$^{-3}$. In March and April, rBC averaged 0.1 µg m$^{-3}$ while decreasing to 0.02 µg m$^{-3}$ in May.

Positive Matrix Factorization (PMF) of the OA mass spectra yielded three factors: (1) a Hydrocarbon-like Organic Aerosol (HOA) factor, which was dominated by primary aerosols and accounted for 12% of OA mass; (2) an Arctic haze Organic Aerosol (AOA) factor; and (3) a more oxygenated Marine Organic Aerosol (MOA) factor. AOA dominated until mid-April (64%-81% of OA), while being nearly absent from the end of May and correlated significantly with $SO_4^{2-}$, suggesting the main part of that factor being secondary OA. The MOA emerged late at the end of March, where it increased with solar radiation and reduced sea ice extent, and dominated OA for the rest of the campaign until the end of May (24-74% of OA), while AOA was nearly absent. The highest O/C ratio (0.95) and S/C ratio (0.011) was found for MOA. Our data supports current understanding that Arctic aerosols are highly influenced by secondary

aerosol formation, and with an important contribution from marine emissions during Arctic spring in remote high Arctic areas. In view of a changing Arctic climate with changing sea-ice extent, biogenic processes, and corresponding source strengths, highly time-resolved data are needed in order to elucidate the components dominating aerosol concentrations to enhance the understanding of the processes taking

place.

**1 Introduction**

Climate change driven by anthropogenic emission of greenhouse gases seriously impacts the Arctic, which has experienced average temperature increases of twice the global mean during the last 100 years (AMAP, 2015; IPCC, 2018). Warming has led to destabilization of permafrost (AMAP, 2017) and a

longer melting season resulting in a critical decrease in the sea-ice extent (Stroeve et al., 2007). The latter changes the Earth's albedo and results in positive sea-ice and snow-albedo feedbacks causing further warming (Lenton, 2012). In addition to long-lived greenhouse gases such as $CO_2$, atmospheric aerosols also have an impact on the radiation balance of the Earth. Aerosols affect the radiative balance in various ways. They can absorb and scatter solar radiation, causing either warming or cooling of the atmosphere,

respectively. Aerosols can also impact the properties of clouds, for example affecting cloud reflectivity, by serving as cloud-condensation and ice nuclei (Twomey, 1977). Due to aerosols' climatic importance it is crucial to expand the knowledge regarding their chemical and physical properties in the Arctic to reduce the current uncertainty (IPCC, 2013) with respect to the overall effect of aerosols on Earth's energy budget.

It is well established that the aerosol concentration in the Arctic atmosphere is seasonally varying resulting in higher loadings during winter and spring, compared to summer and fall, often referred to as "Arctic haze" (Heidam et al., 2004; Tunved et al., 2013; Heidam et al., 1999; Quinn et al., 2007; Barrie et al., 1981; Heidam, 1984). This is explained by a greater accessibility to the lower troposphere in the Arctic from anthropogenic source regions outside the Arctic due to an expansion of the polar dome

(AMAP, 2011) in winter and spring. In addition, during the Arctic winter strong temperature inversions create stable stratification where aerosol removal processes are strongly reduced prolonging their atmospheric lifetime (Stohl, 2006; Sodemann et al., 2011; AMAP, 2011). The air masses inside the wintertime dome are extremely dry, limiting aerosol wet deposition, while low turbulence caused by the stratification and slow vertical exchange reduces the dry deposition of aerosols (Sodemann et al., 2011;

Stohl, 2006; Abbatt et al., 2019). The Arctic haze peaks in early spring (Heidam et al., 1999; Law and Stohl, 2007; Stohl, 2006; Heidam et al., 2004; Abbatt et al., 2019). Arctic haze particles effectively scatter light (Andrews et al., 2011; Schmeisser et al., 2018), and act as cloud condensation nuclei (CCN) (Earle et al., 2011; Komppula et al., 2005). Due to the expansion of the polar dome, a major part of the aerosol mass is long-range transported from source regions outside the Arctic where the primary source region

has been identified as the northern part of Eurasia (Nguyen et al., 2013; Quinn et al., 2008; Heidam et al., 2004; Stohl et al., 2007; Christensen, 1997; Abbatt et al., 2019). Studies have shown that main constituents of Arctic aerosols are sulfate ($SO_4^{2-}$) and organics mixed with a minor fraction of nitrate ($NO_3^-$), ammonium ($NH_4^+$), black carbon (BC) and heavy metals (Quinn et al., 2007; Fenger et al., 2013;

Nguyen et al., 2013; Frossard et al., 2011; Barrie et al., 1981). This is also the case at the high Arctic station, Villum Research Station (VRS) at Station Nord in North Greenland, where this study was conducted. Rahn and Heidam (1981) have previously estimated the average chemical composition of Arctic sub-micrometer aerosols during winter-spring to 2 µg m$^{-3}$ SO$_4^{2-}$, 1 µg m$^{-3}$ organic aerosol (OA), 0.3-0.5 µg m$^{-3}$ BC and a few hundred ng m$^{-3}$ of other compounds. Since then, SO$_4^{2-}$ and BC during winter-spring have declined at Alert, Mount Zeppelin, Barrow and VRS (Heidam et al., 1999; Hirdman et al., 2010; AMAP, 2015). However, the total Arctic column burden may have increased (Sharma et al., 2013).

BC is the most important aerosol at absorbing solar radiation in the atmosphere. Of particular concern for the Arctic, BC deposited on snow and ice-covered surfaces changes the albedo, leading to increased absorption of solar radiation and direct heating of the surface (Bond et al., 2013). Consequently, melting accelerates giving BC an important role especially in an Arctic context (Bond et al., 2013; Quinn et al., 2008; AMAP, 2011). Long-range transport of BC to the Arctic is very effective in mid-winter, when removal processes are slowest. Transport reaches a minimum in late spring where wet deposition becomes an important removal process (Abbatt et al., 2019; AMAP, 2015). Natural emissions from vegetation fires can be considerable in spring and early summer (Mahmood et al., 2016). Overall, the general seasonal cycle of BC in the Arctic is characterized by highest concentrations observed between January and April and lowest concentrations throughout the summer, but with periodic spikes in concentration throughout the summer (Sharma et al., 2006). OA is also an important component of Arctic aerosols and is composed of many different molecules derived from either primary emissions or from secondary production. Consequently, there are often many distinct sources of OA. OA can typically contribute up to one third of PM$_1$ in the Arctic though few studies have characterized this component in detail (Barrett et al., 2015; Brock et al., 2011; Frossard et al., 2011; Kawamura et al., 2010; Quinn et al., 2002; Shaw et al., 2010; Leaitch et al., 2018; Chang et al., 2011; Willis et al., 2018). Total OA is relatively constant or decreasing with time in late winter. However, during spring it increases suggesting that there is photochemical production of OA (Willis et al., 2018). There is a need for more detailed measurements of OA composition in the Arctic to better understand the key sources and how these vary with time (Willis et al., 2018).

It is crucial to understand natural sources in addition to anthropogenic sources of Arctic aerosols. Marine and coastal marine locations constitute a large part of Arctic, and marine aerosols comprise both organic and inorganic constituents of primary and secondary origin. Production of primary marine aerosols is known to correlate with wind speed and possibly also other mechanisms (Willis et al., 2018). Primary marine organic aerosols in Arctic regions are believed to consist of water soluble or surface-active organic compounds present in the surface water, or water insoluble microgels (Willis et al., 2018; Leck and Bigg, 2005; Orellana et al., 2011). Marine aerosols play an important role for the climate due to their optical properties and ability to alter cloud nucleation (Abbatt et al., 2019; Willis et al., 2018). Biogenic marine aerosols can scatter solar radiation, which will result in a negative radiative forcing. Biogenic marine aerosols can also coat soot particles, which may be transported from wild fires (AMAP, 2015), which could impact the CCN activity and absorption by the soot particles (Lange et al., 2018). Methane

sulfonic acid (MSA), an oxidation product of dimethyl sulfide (DMS) is abundant in spring and summer (Abbatt et al., 2019) and is a key indicator of secondary marine aerosols. MSA levels have been associated with marginal sea ice moving North (Laing et al., 2013; Quinn et al., 2009; Sharma et al., 2012). A new satellite-based model suggests that DMS emissions in the Arctic have increased by 30% per decade the last two decades due to both increased temperatures and decreased ice cover (Abbatt et al., 2019). A relationship between MSA and the frequency of new particle formation has also been inferred based on long-term observations (Dall'Osto et al., 2017) although MSA cannot be the nucleating part. This suggest that DMS is important for summertime particles. Another important natural source of Arctic aerosols is ammonia, which among other things is believed to originate from migrating sea bird colonies (Croft et al., 2016). Modeling studies have been shown to better capture particle burst and growth when an ammonia source from sea birds were included (Croft et al., 2018; Croft et al., 2016). Additionally, ammonia can also be transported from boreal wildfires from lower latitudes.

Many previous Arctic studies have been based on off-line analysis and filter measurements of ambient aerosols with a relatively low time resolution of hours up to a week (Heidam et al., 1999; Heidam et al., 2004; Skov et al., 2006; Quinn et al., 2007; Massling et al., 2015; Leaitch et al., 2018; Sharma et al., 2012; Quinn et al., 2009). Beside the low time resolution, a disadvantage of these types of measurements can be evaporate loss or adsorption of semi-volatile compounds (Lee et al., 2013; Dillner et al., 2009). Highly time-resolved in-situ measurements can reduce these artifacts while also enabling the possibility to observe the variations of different chemical species on a much shorter time-scale. In this way, it is possible to look into the processes behind the observed levels. In the last decade, Aerosol Mass Spectrometry (AMS) (Canagaratna et al., 2007; DeCarlo et al., 2006; Jimenez et al., 2003; Drewnick et al., 2005; Jayne et al., 2000) has been widely used as an on-line method for quantitative analysis of chemical composition of atmospheric particles. With the addition of a laser vaporizer (Onasch et al., 2012), its application has been extended to include refractory aerosol components, including refractory black carbon (rBC).

In this study, the time dependent concentrations of sub-micrometer particle composition including OA, $SO_4^{2-}$, $NO_3^-$, $NH_4^+$, chloride ($Cl^-$) and rBC are reported at the high Arctic site VRS. The measurements were conducted by application of a soot particle aerosol mass spectrometer (SP-AMS) and auxiliary measurements during the Arctic spring 2015, when concentrations are expected to peak. The objectives are to gain better insight into the processes influencing the chemical composition of high Arctic aerosols and to allocate potential sources and source types by use of positive matrix factorization (PMF).

## 2 Experimental

### 2.1 Sampling site

The atmospheric measurements were carried out at VRS located at the Danish military station, Station Nord in North Greenland (Figure S1, $81^0$ 36'N, $16^0$ 40'W, 24 m above mean sea level). VRS is situated in a region with a dry and cold climate where the annual precipitation is 188 mm and the annual mean temperature is -21 °C. The dominating wind direction is southwestern with an average wind speed of 4

m s$^{-1}$ as apparent from Figure S1 (Rasch et al., 2016; Nguyen et al., 2013). The SP-AMS data were sampled in an atmospheric observatory containing two laboratories whereas data from a multi-angle absorption photometer (MAAP) and a filter pack sampler was collected in a smaller co-located hut (Flygers hut) - both equipped with particle and gas inlets. The two measurement sites are located 2.5 km southeast of the military station and are only 300 meters apart. Given the close proximity of the two

laboratories and the lack of hyper-local sources, we expect both to sample largely the same air mass. A high-volume sampler (HVS) provided filter samples for off-line analysis. The HVS was located at the outskirts of the military station, hence 2.5 km from the main sampling site. More information concerning the supplementary instruments can be found in Supporting Information. All particulate measurements in the Atmospheric Observatory were conducted by drawing air through a slightly heated (absolute 5 °C)

particle inlet custom-built by TROPOS (Leipzig, Germany). Sampling took place during a CRAICC (Cryosphere-Atmosphere Interactions in a Changing Arctic Climate) field campaign from 20 February until 23 May 2015.

**2.2 The soot-particle aerosol mass spectrometer**

An SP-AMS (Aerodyne Research Inc.) was deployed at VRS for measuring mass concentration and

chemical composition of sub-micrometer aerosols with a time resolution of two minutes. The SP-AMS is described in detail elsewhere (Onasch et al., 2012). In brief, the instrument samples aerosols into a vacuum chamber through an aerodynamic particle lens, which creates a narrow particle beam. In the vacuum chamber, the aerosols accelerate to a velocity depending on their vacuum aerodynamic diameter enabling analysis of the aerosol size distribution. Subsequently, the aerosols undergo vaporization,

ionization with 70 eV electron impact, and detection with time-of-flight mass spectrometry. The vaporization of aerosols components in the SP-AMS can occur in two ways: (1) impaction on a tungsten surface at a temperature of 600 °C, or (2) intersection with the beam of a continuous-wave 1064 nm intracavity Nd:YAG laser. The laser extends the application of the AMS to include refractory particulate matter (R-PM) since it enables vaporization of strongly infrared light absorbing particles, such as

refractory BC (Onasch et al., 2012). In this study, high-resolution (HR) mass concentrations of $SO_4^{2-}$, $NO_3^-$, $NH_4^+$, organics, $Cl^-$ and rBC are obtained from the SP-AMS.

The SP-AMS was operated in two minutes laser off and two minutes laser on in V-mode and alternated between the mass spectrum mode and the particle time-of-flight (pToF) to obtain sub-micrometer particles (PM$_1$). Non-refractory species are reported for time periods where the laser was off. The flow

rate was controlled regularly with a Gilian Gilibrator (Sensidyne). During the first part of the campaign, ionization efficiency (IE) calibrations with ammonium nitrate particles were conducted on a weekly basis and during the last part every second week. To establish the detection limit and to enable adjustments of the fragmentation tables a high-efficiency particulate air (HEPA) filter was applied on a daily basis for a period of 30 to 60 minutes with a time resolution of 2 minutes. The lower detection limit of the different

species was determined as three times the standard deviation of the mass concentration during the HEPA filter periods (Table 1). The data were analyzed with the standard AMS Igor Pro-based (version 6.35 Wavemetrics, Inc) software tools SQUIRREL (version 1.57G) and PIKA (version 1.16H), available at

http://cires1.colorado.edu/jimenez-group/ToFAMSResources/ToFSoftware/index.html. The analysis followed the principles described in DeCarlo et al. (2006), Jimenez et al. (2003); Allan et al. (2004) and Onasch et al. (2012).

The default relative ionization efficiency (RIE) values for OA, $SO_4^{2-}$, $NO_3^-$ and $Cl^-$ of 1.4, 1.2, 1.1 and 1.3, respectively, were applied, which are based on Canagaratna et al. (2007). A RIE of 3.5 was applied for $NH_4^+$. It should be noted that chloride reported in the current study is measured with laser off and is thus non-refractory chloride and largely excludes refractory species such as chloride in sea salt aerosols. Thus, reported $Cl^-$ in this study is most likely primarily a sum of organic $Cl^-$ and $NH_4Cl$ due to the acidic environment at VRS. However, the partitioning of chloride between different species has not been investigated further, since it is not within the scope of this study. A RIE for rBC of 0.46 was found from calibrations with Regal Black (a commercial carbon black). The appropriateness of this RIE for ambient Arctic rBC is discussed in Section 2.4. Calibrations with Regal Black and ammonium nitrate were done with the same frequency. Fragment ions from organic species can overlap with some of the marker ions for rBC. To minimize the organic contribution to the nominal rBC signal (especially at $C_1^+$ an organic contribution was evident), $C_3^+$ was used to quantify rBC. Thus, the $C_3^+$ signal was scaled with a factor of 1/0.55 to match the fraction in the Regal Black mass spectra (Martinsson et al., 2015). The applied collection efficiency (CE) for non-refractory PM and rBC will be discussed in more detail in a subsequent section.

## 2.3 Auxiliary equipment

The aerosol light absorption was measured using a MAAP (Model 5012 Thermo Scientific) operated at a flow rate of 1 $m^3$ $hour^{-1}$ with an inlet without a size cut-off. Aerosols were sampled on a filter in which the light absorption at 670 nm was measured by a photometer. Detailed information about the instrument can be found in Petzold and Schonlinner (2004) and previous MAAP measurements from VRS are published in Massling et al. (2015). The BC concentration is determined from the relationship between the aerosol light absorption coefficient and a specific aerosol absorption coefficient (Petzold and Schonlinner, 2004). The specific absorption coefficient describes BCs ability to absorb solar radiation at a specific wavelength, which depends on the age of the aerosol (Petzold et al., 1997; Sharma et al., 2002) and is often determined based on correlations with thermal-optical measurements of elemental carbon (EC) (Sharma et al., 2004). In this study, the MAAP's default value of 6.6 $m^2$ $g^{-1}$ has been applied based on Massling et al. (2015). Uncertainty in the conversion factor likely impacts the reported absolute concentrations, and potentially the temporal variability. In addition, a scanning mobility particle sizer (SMPS) measured the particle number size distribution, which was used for validating the SP-AMS results. The SMPS is custom-built with a Vienna-type medium column and more information can be found in Lange et al. (2018). A description of the validation can be found in Supporting Information.

## 2.4 Comparison between instruments

A collection efficiency (CE) adjustment is normally applied to AMS data, which accounts for particle loss in the instrument caused by the inlet and the aerodynamic lens, beam divergence, and particle bounce

effects (Canagaratna et al., 2007; Onasch et al., 2012). In this study, the parameterization developed by Middlebrook et al. (2012) has been used where a time dependent CE is determined based on the aerosols chemical composition. Previous studies have shown an increasing CE with particle acidity, the content of nitrate, and relative humidity (Quinn et al., 2006; Jayne et al., 2000; Matthew et al., 2008). The time dependent CE varied with the majority (> 97%) of values between 0.8 and 1 (Figure S2). In this study, the high CE was due to acidic aerosols. This is also evident from Figure S3.a showing that the theoretical predicted $NH_4^+$ concentration necessary for neutralizing the mass concentration of inorganic anions is much larger than the actual $NH_4^+$ concentration measured by the SP-AMS (slope = 0.14). The acidity is explained by the high amount of sulfuric acid.

Applying the RIE for rBC of 0.46 determined from Regal Black calibrations, a good correlation between rBC and $BC_{MAAP}$ is found (Figure S3.b). While there is a strong linear relationship between the two ($R^2$ = 0.83), the $BC_{MAAP}$ was about three times larger than the SP-AMS rBC (slope = 0.33 ± 0.02). This indicates that the actual RIE for rBC was lower than the value of 0.46 determined during laboratory calibrations. A lower RIE can be explained by different particle size and a more complex morphology of the Arctic soot compared to the Regal Black used for calibration. An effective RIE is determined for rBC by forcing the SP-AMS measurements to match the MAAP measurements. For rBC an effective RIE of 0.15 (= 0.33 * 0.46) is hence applied in this study.

Comparison of the total $PM_1$ mass concentration (sum of OA, $SO_4^{2-}$, $NH_4^+$, $NO_3^-$, $Cl^-$ and rBC) with the calculated total volume from the SMPS assuming spherical particles was carried out to validate the SP-AMS results. The SMPS was operated to characterize particles having mobility diameters between 9 and 870 nm. This corresponds to a larger size range than sampled by the SP-AMS, which has 100 % transmission efficiency within aerodynamic diameters between 70 and 600 nm, and adjustment from aerodynamic diameter to mobility diameter further brings the SP-AMS into the SMPS range (DeCarlo et al., 2006; Allan et al., 2003). However, previous studies (Nguyen et al., 2016; Lange et al., 2018) have shown that the dominant particle size range at VRS during winter and spring months is within detection range of the SP-AMS. Thus, the number of particles from the SMPS exceeding the size range measured by the SP-AMS should be relatively small and thereby not influence the results, since particles in the lower end of the size distribution do not significantly contribute to volume. There was a generally reasonable temporal correspondence between the two measurements. Although there were some periods where they differed notably it were within the expected range given the accuracy of the two instruments. A more detailed discussion about the comparison between the two instruments is presented in Supporting Information (Figure S5).

**2.5 Positive Matrix Factorization**

PMF analysis (Paatero, 1997; Paatero and Tapper, 1994; Lanz et al., 2007; Ulbrich et al., 2009) was conducted on the time dependent organic mass spectra to determine OA factors and potential sources of OA. The analysis was carried out with the PMF Evaluation Tool Software (PET, v2.08D; available online at http://cires1.colorado.edu/jimenez-group/wiki/index.php/PMF-AMS_Analysis_Guide) on mass spectra consisting of HR ions with $m/z$ values from 12 to 100. The detailed procedure is described

elsewhere (Ulbrich et al., 2009; Zhang et al., 2011). The input HR mass spectra and error matrix with the appropriate ion fragments were generated in PIKA, where the error matrix was calculated as the sum of the quadrature of the electronic noise and Poisson counting for each ion (Allan et al., 2003). Isotopes were removed from both the data and error matrix since they would give additional weight to the parent ion in the PMF analysis.

As described in Ulbrich et al. (2009) "weak" ions with a signal-to-noise ratio (SNR) between 0.2 and 2 were down-weighted by a factor of 2 whereas "bad" ions with a SNR below 0.2 were removed from the data and error matrix. The PMF was executed in exploration mode with a range of factors (between 1 and 5). The robustness of the solutions was tested by setting different random starting points (SEED: 0 to 10, steps = 1) (Zhang et al., 2011). The detailed procedures for choosing the best solution were based on Zhang et al. (2011). A solution with three factors (Figure 2) was identified after evaluating $Q/Q_{exp}$ and residuals, interpreting the mass spectra and investigating the temporal correlation between the factor time series and potential tracer species (Ulbrich et al., 2009; Zhang et al., 2011). FPEAK and seed values were changed to test the stability of the three-factor solution and based on the diagnostic plots a three-factor solution was selected with a FPEAK and seed value of zero (Figure S7). A 4-factor solution was scientifically not meaningful with respect to the chemical composition and returned an O/C ratio >> 1 for one of the factors. Hence, we do not observe a fourth "continental" factor, which has been previously observed during the ASCOS cruise track in the summer/autumn season around Svalbard (Chang et al., 2011). If present, the continental factor is most likely of negligible abundance for which reason the PMF-analysis cannot differentiate it from other oxygenated organic aerosol (OOA). Detailed information regarding the factor combination can be found in Supporting Information.

## 3 Results and Discussion

### 3.1 Time series

Time dependent OA, $SO_4^{2-}$, $NO_3^-$, $NH_4^+$, $Cl^-$ and rBC concentrations [µg m$^{-3}$] measured by the SP-AMS are presented in Figure 1 together with temperature [°C], mean wind speed [m/s], and wind direction [°] for the time period 21 February to 23 May 2015. Weekly average concentrations can be found in Figure S6. Figure 1c shows the time dependent mass fraction of the different species. The total measured $PM_1$ concentration during the field study may seem relatively high, averaging 2.3 µg m$^{-3}$- ranging from 2.3, 2.3 and 3.3 µg m$^{-3}$ in February, March and April to 1.2 µg m$^{-3}$ in May. It should be emphasized that this average does not consider particulate water, NaCl, and elements such as K, Ca, Si, Al and Fe. These elements may additionally contribute 0.1 – 0.2 µg m$^{-3}$ to $PM_1$ (Nguyen et al., 2013; Heidam et al., 2004). The measurement period covers the Arctic late winter and spring where high aerosol loadings are expected due to the favorable conditions for long-range transport of aerosols from mid-latitudes and slow particle removal rates. With regard to $PM_1$ concentration we hence observe the typical Arctic haze phenomenon. Generally, the area around VRS is dominated by winds from southwest (Nguyen et al., 2013), which is also evident during this campaign (Figure S1). As expected no diurnal pattern is observed for any of the chemical species. These are mainly transported from long distances. For example, the

source regions that contributed to ground-level $SO_X$ at VRS were assigned to Western Europe (7%), Eastern Europe (9%), Asia (2%), North America (7%) and Russia being the main emitter by far (75%) (Heidam et al., 2004). During summer, the atmospheric circulation is confined within the Arctic region and is considered essentially local. Thus, marine biogenic sources that peak during spring and summer are expected to origin from within the region. Arctic sites show similar increases in key particulate pollutants in winter and early spring, where maximum sulfate concentrations may reach 3 $\mu g\ m^{-3}$ as compared to average summer concentrations of 0.1 $\mu g\ m^{-3}$ (Quinn et al., 2007). For example, typical $PM_1$ concentrations were 0.1 - 0.2 $\mu g\ m^{-3}$ in August to September during the ASCOS expedition (Chang et al., 2011). Sulfate is dominated by anthropogenic sources accounting for 65% at Alert (Norman et al., 1999) and 75% Svalbard (Udisti et al., 2016) as annual averages. On the contrary, biogenic sources accounted for 63% of sulfate in size fraction smaller than 490 nm at Alert during summer (Ghahremaninezhad et al., 2016).

During the entire campaign, $SO_4^{2-}$ is the dominant species that on average makes up almost 70% of the $PM_1$ mass concentration with highest concentration until the end of April and decreasing in May (Figure 1b-c). This is in accordance with previous findings for $SO_4^{2-}$ at VRS based on measurements with lower time-resolution (Nguyen et al., 2013; Fenger et al., 2013; Heidam et al., 2004). Atmospheric $SO_4^{2-}$ is mainly formed as secondary inorganic and only a minor fraction is from primary emissions (Massling et al., 2015). Secondary $SO_4^{2-}$ is formed by atmospheric oxidation of sulfur dioxide ($SO_2$) and to some extent DMS (as the long-range transport is occurring over sea ice), and is dependent on the oxidative capacity of the atmosphere e.g. the concentration of hydroxyl radicals (OH). Secondary long-range transported $SO_4^{2-}$ depends on atmospheric oxidation of $SO_2$ at the vicinity of the source regions, whereas local transformation (close to VRS) of $SO_2$ leads to higher concentration of $SO_4^{2-}$ from March, where solar radiation is sufficient with peak radiation exceeding 100 $W/m^2$ (Figure 3). This is consistent with results reported from other Arctic sites (Quinn et al., 2007; Gong et al., 2010; Heidam et al., 2004; Skov et al., 2017). Previous studies suggest that the main source of $SO_2$ and $SO_4^{2-}$ at VRS is long-range transport of anthropogenic emissions mainly originating from Siberia (Heidam et al., 2004; Nguyen et al., 2013). In winter and early spring, direct emissions of sea-salt sulfate and photo-oxidation of oceanic emissions of DMS were expected to play a minor role since the ocean surrounding VRS is frozen at that time of year (Heidam et al., 2004). However, a recent study using both airplane measurements and modeling suggest that long-range transport of DMS is significant during spring (Ghahremaninezhad et al., 2017). From the beginning of April, the sea ice extent of the Northern Hemisphere is markedly reduced, and at the same time solar radiation increases (Figure 3). In this period, we observe MSA as an ion in the SP-AMS at $m/z$ 78.9854. MSA is formed by atmospheric oxidation of DMS, which results from bacterial breakdown of dimethylsulfoniopropionate produced by marine phytoplankton and microalgae (Carpenter et al., 2012). In this study, MSA emerges steadily and peaks the end of April (see Section 3.2). Oxidation of DMS may involve the hydroxyl radical, ozone, and halogen radicals such as $Cl^-$ and BrO (Barnes et al., 2006; Hoffmann et al., 2016).

In this study, the OA fraction is the second largest contributor to $PM_1$ where weekly averages showed a clear decrease from mid-April relative to concentrations in February and March (Figure 1). The OA time

dependent concentration shows relatively large peaks during shorter time periods, which in some cases

can be attributed to a change in wind direction from Southwesterly to Northerly winds (around 10°, Figure S1). While these wind directions were registered on a few occasions they potentially provided local pollution from the military station located three kilometers away from the measurement site. These peaks have not been discarded and the impacts of local pollution will be discussed further in Section 3.2.

Particulate $NH_4^+$ is found in much lower concentrations compared to OA and $SO_4^{2-}$ but with the same

transition pattern as the two other species. For the campaign, a significant correlation is found between $SO_4^{2-}$ and $NH_4^+$. However, it is known that $SO_4^{2-}$ and $NH_4^+$ do not originate from the same sources. $SO_2$, a key precursor to $SO_4^{2-}$, originates from combustion of fossil fuel and is oxidized to $SO_4^{2-}$ in the atmosphere. In contrast, ammonia ($NH_3$) which is the precursor of $NH_4^+$, derives largely in winter and spring from long-range transport of emissions from biomass burning and agriculture (Fisher et al., 2011),

whereas in summertime $NH_3$ emission from seabird-colonies can play a significant role (Croft et al., 2016). The strong correlation between $SO_4^{2-}$ and $NH_4^+$ ($R^2 = 0.70$) suggests that the acidity of the particles is reasonably constant with time. This is furthermore in agreement with the general assumption that $NH_4^+$ is bound irreversibly to $SO_4^{2-}$ (e.g. Seinfeld and Pandis, 1998), in this case as ammonium bisulfate. Particle bound $NH_4^+$ has a much longer lifetime than $NH_3$ (Baek and Aneja, 2004) and therefore it is

transported as $NH_4^+$ even to the high Arctic.

The average concentration of $NO_3^-$ and $Cl^-$ are 0.03 and 0.02 μg m$^{-3}$, respectively, which is close to the detection limits. These concentration levels are lower compared to what has previously been observed at VRS (Fenger et al., 2013; Heidam et al., 2004). However, the SP-AMS does not typically measure refractory chloride at normal vaporizer temperatures, such as NaCl (Canagaratna et al., 2007). Although,

Ovadnevaite et al. (2012) has demonstrated how the AMS could be calibrated to measure NaCl in high-time resolution. Moreover, Fenger et al. (2013) found that the overall size distribution of chloride and $NO_3^-$ differed from $SO_4^{2-}$, with $Cl^-$ and $NO_3^-$ mainly found in supermicrometer particles (> 1 μm) not detectable by SP-AMS. Based on the size of the particles and air mass back-trajectories Fenger et al. (2013) suggested that the particles originate from local/regional sources (frost flowers and refreezing

leads). Only during certain periods with specific wind directions $NO_3^-$ and $Cl^-$ were found in accumulation mode particles, which were ascribed to long-range transported particles (Fenger et al., 2013). Current research has suggested that blowing snow might be a much more dominant source of sea salt aerosols compared to frost flowers (Huang and Jaegle, 2017).

The highest rBC loadings are found in the first month of the campaign (February) averaging 0.2 μg m$^{-3}$.

In March and April, the average is 0.1 μg m$^{-3}$ which then decreases to 0.02 μg m$^{-3}$ in May. As with OA, some of the spikes in the rBC time series are related to a change in wind direction and likely the result of local pollution from the military station. All data are included here and missing time periods of rBC (during April and May) are due to technical problems with the SP-AMS laser. BC is primarily emitted from both anthropogenic and natural combustion sources (Bond et al., 2013). Upon emission, aerosols

containing BC grow by condensation and coagulation into the accumulation mode. These accumulation mode BC-containing particles can be transported over longer distances during the Arctic haze period and may serve as cloud seeds in the late spring, when precipitation begins to be important in the Arctic (Bond

et al., 2013; AMAP, 2011; Massling et al., 2015; Garrett et al., 2011). Further, condensational growth of the BC-containing particles may increase the absorption by these particles (Cappa et al., 2012; Liu et al., 2015). Previous studies have found a correlation between BC and $SO_4^{2-}$ at different Arctic stations (Massling et al., 2015; Eckhardt et al., 2015; Hirdman et al., 2010). These studies suggest that the two species are internally mixed and possibly undergo similar transport patterns. Furthermore, comparable correlation slopes were found for the different Arctic locations, which suggest that source regions of BC and $SO_4^{2-}$ could be similar throughout the Arctic. An even more recent study suggests that only a minor part of ambient aerosols contained rBC inclusions (Kodros et al., 2018). We find a significant correlation between the two species (students t-test, level of significance 99.995), consistent with previous studies. However, we also find that the $R^2$ value is relatively low (0.18). The reason for this is that there are periods with particularly high rBC concentrations, likely originating from local emission sources (e.g. the military base), which will be investigated further in the following section. Additionally, in April and May $SO_4^{2-}$ from DMS oxidation will make up a larger fraction of total $SO_4^{2-}$, and thereby reduce the ratio between rBC and $SO_4^{2-}$, which is also evident from Figure S4.

## 3.2 Source Apportionment

The PMF analysis was conducted for the HR OA mass spectra with one to five PMF factors and a three-factor solution was chosen (more details can be found in Supporting Information). Figure 2 shows the mass spectral profiles of the three different factors for the entire campaign period. Figure 3 illustrates time series for the factors and Table 2 shows the correlation of each factor with tracer species, respectively. Figure 4 illustrates the average mass concentration ($\mu g\ m^{-3}$) and the mass fraction of the factors in February, March, April and May. The PMF analysis yielded three factors: 1) a hydrocarbon-like organic aerosol factor (HOA), 2) an oxygenated Arctic haze organic aerosol factor (AOA) dominating winter and early spring, and 3) a more oxygenated marine organic aerosol factor (MOA) which builds up in late spring and becomes the dominating OA throughout late spring. The identification of these factors is discussed below.

The HOA factor is characterized by hydrocarbon fragments especially at *m/z* 41, 43, 55, 57, 67, 69 and 71 ($C_3H_5^+$, $C_3H_7^+$, $C_4H_7^+$, $C_4H_9^+$, $C_5H_7^+$, $C_5H_9^+$, $C_5H_{11}^+$, respectively) from chemically reduced organic emissions. The O/C ratio of 0.11, high signal at *m/z* 57 and the absence of $CO_2^+$ is a characteristic of primary combustion sources of fossil origin, which is similar to other HOA factors found in previous studies (Zhang et al., 2005; Aiken et al., 2009) and at other Arctic locations (Frossard et al., 2011). The very small contribution from the $CO_2^+$ at *m/z* = 44 and the very small abundances of typical biomass burning OA (BBOA) marker ions at *m/z* 60 ($C_2H_4O_2^+$) and *m/z* 73 ($C_3H_5O_2^+$) in the HOA factor spectrum suggests that the HOA factor is not mixed with BBOA. This finding is consistent with previous results that indicate BBOA levels are typically very low, based on measurements of levoglucosan in the Arctic, (Zangrando et al., 2013). The time series of HOA and rBC showed a moderate correlation ($R^2 = 0.35$), which is consistent with the HOA factor being of primary origin. The relatively low $R^2$ value (Table 2) can be partly explained by rBC being internally mixed with $SO_4^{2-}$ and transported with the AOA factor. The HOA time series is generally higher in concentration at the beginning of the measurement period

(Figure 4). The time series of HOA reveals a number of shorter periods with high mass loading, which could be caused by local pollution from the military station 2 km north of the measurement site due to a change in wind direction, or exhaust plumes from snow scooters and heavy-duty vehicles occasionally clearing the road nearby the measurement station for snow (see windrose, Figure S1). It is not trivial to distinguish local events and, in this case, the possible local contamination was investigated by comparing high HOA peaks ($> 0.45$ µg m$^{-3}$) with size distribution measurements from the SMPS (Lange et al., 2018). Periods which were attributed to local contamination accounted for less than 1% of OA concentration. Therefore, essentially the entire HOA concentration is assigned to long-range transportation, possibly sources with different ratios of HOA and rBC which would explain the moderate correlation between HOA and rBC.

Oxygenated aerosols from numerous field campaigns on the northern hemisphere are deconvolved into HOA and OOA. OOA has been shown to account for a large fraction of OA and to be a good surrogate for secondary organic aerosols (SOA) in multiple studies (Ng et al., 2010; Zhang et al., 2007; Zhang et al., 2011). Oxygen containing functional groups produce $m/z$ 43 ($C_2H_3O^+$) and $m/z$ 44 ($CO_2^+$) fragments, which are prominent peaks in OOA mass spectra (Ng et al., 2010), including those of MOA and AOA found in this study. These factors are highly OOA factors with O/C ratios of 0.63 and 0.95, respectively. According to Jimenez et al. (2009) these factors would be classified as low volatility OOA (LV-OOA). There is strong evidence that OOA is secondary in nature and several studies of aging indicate that OA converges towards LV-OOA following numerous steps of atmospheric oxidation (Jimenez et al., 2009). The AOA is the most abundant factor from the beginning of the campaign through mid-April. AOA accounts for 64% of OA mass for the entire field study but ranges from 64%, 81% and 71% of OA in February, March and April to 20% in May (Figure 2b and 4). The dominating OA appears to origin from long-range transport into the region during winter/spring. At the end of April and onwards the factor was nearly absent, which is in agreement with increasing wet deposition in the spring and a contracting polar dome impairing long-range transport into North Greenland (Abbatt et al., 2019). Generally, an OOA factor mainly consists of SOA but can also include oxygenated organic species from primary emissions (Zhang et al., 2005). In this case the AOA factor correlates significantly (level 99.995) with $SO_4^{2-}$, which is mainly formed by atmospheric oxidation of $SO_2$ suggesting the main part of the factor being SOA. The correlation is especially good until mid-April after which $SO_4^{2-}$ begins to correlate with MOA. The O/C ratio of 0.63 also indicates a less oxidized and fresher SOA factor, or an SOA formed from generally larger precursor volatile organic compounds (VOCs), similar to what has been found in previous studies (O/C between 0.52 – 0.64, (Aiken et al., 2008)). The AOA mass spectrum also included mass spectral peaks at $m/z$ 60.021 ($C_2H_4O_2^+$) and 73.029 ($C_3H_5O_2^+$). These fragments are often taken as being indicative of anhydrous sugar such as levoglucosan, and thereby suggest that biomass burning makes some contribution to Arctic OA. However, SOA also contributed to the abundance of $C_2H_4O_2^+$ (Aiken et al., 2008; Aiken et al., 2009; Cubison et al., 2011; Lee et al., 2010; Saarnio et al., 2013). Quantitatively, the expected abundance of $C_2H_4O_2^+$ from SOA did not exceed the measured concentration in this study. Biomass burning is generally assumed to play a significant role in the context of the composition of the Arctic aerosol (Stohl et al., 2013) where recent publication using isotopes of carbon reports biomass

burning or biofuel use to account for up to 57% of EC at the Arctic station Zeppelin at Svalbard during high pollution events in winter (Winiger et al., 2015). However, levoglucosan is prone to atmospheric oxidation by hydroxide radicals (OH) (Hennigan et al., 2010; Hoffmann et al., 2010), which could degrade the markers during transport to North Greenland. This can explain the low abundance of levoglucosan markers measured in this study.

The MOA factor has a mass spectrum dominated by $m/z$ 28 and 44 ($CO^+$ and $CO_2^+$), of which the latter is probably a fragment from e.g. organic acids and acid derived species, such as esters (Duplissy et al., 2011). An O/C of 0.95 reveals that the factor is highly oxidized and most likely photochemically aged. The MOA spectrum resembles a marine organic plume previously published from Mace Head, in the North East Atlantic Ocean showing evidence of both primary and secondary organic aerosols of marine origin (Ovadnevaite et al., 2011). Most abundant peaks in this spectrum were oxygenated fragments at $m/z$ 28 and 44. Also prominent were $m/z$ 27, 39 and 41 from the CH family, and $m/z$ 43 and 55 from the CHO family, which are also found in the MOA spectrum. The two spectra differ in terms of abundances of CH-like organic matter, but they are different from the marine organic aerosol factor published during the ASCOS expedition in the Central Arctic Ocean (Chang et al., 2011), which shows a closer resemblance with the mass spectrum of pure MSA, i.e. dominating peaks at $m/z$ 15, 48, 64 and 79. The distinct peak at $m/z$ 78.9854 is specific for MSA (Huang et al., 2017), and reveals that MOA has a secondary biogenic source (Becagli et al., 2013). The resemblance of MOA from this study with the mass spectrum from Mace Head and the high O/C ratio of 0.95 indicate, that MOA is composed of chemically aged aerosols from both oxidation of primary aerosols and secondary organic aerosols (Ovadnevaite et al., 2011; Fu et al., 2015). Aerosol growth has been correlated with the presence of MSA, and other organic species (Willis et al., 2016).

Figure 3 and 4 illustrate HOA and AOA decreasing around mid-April, while MOA builds up from the end of March. In 2015, Arctic sunrise onset at February 28[th] at VRS, where the sun became visible for a few minutes. Polar daytime initiates photochemistry and hence the production of OH radicals (Seinfeld and Pandis, 2006) and reactive halogen radicals (Hoffmann et al., 2016; Barnes et al., 2006). From mid-April, the sun is above the horizon all day until the beginning of September. Still solar radiation varies over the day and hence the OH production. In contrast, the concentration of OH during buildup of Arctic haze is correspondingly low with ozone being the major oxidant during the dark winter. In Figure 3, the daily averaged solar radiation (W m$^{-2}$) and sea ice extent (km$^2$) on the Northern Hemisphere are shown together with the time series of MOA. While MOA is less abundant during February and March, this factor greatly increases in April, when radiation exceeds approximately 100 W m$^{-2}$. In April, the highest OA concentrations is observed where AOA accounts for around 70% of OA (Figure 4). In May, MOA becomes the dominating OA while AOA nearly disappears. At the same time, we observe the lowest concentration of OA (0.01 μg m$^{-3}$) consisting of 75% MOA (Figure 4). This is significantly higher than observed at Alert by Narukawa et al. (2008) where marine organic matter contributed 45% to aerosol total carbon in late spring (26 April – 6 May 2000). However, direct comparison is difficult due to different methods and time periods (Narukawa et al., 2008). Until the beginning of April, the sea ice extent is constant at around 14.5 million km$^2$ on the Northern Hemisphere (Figure 3). Hereafter, about a

month after the onset of polar daytime, the sea ice surface area starts to decline. After 6 weeks starting from a constant sea ice extent in mid-May, it is reduced by 2 million $km^2$ corresponding to a 14% loss of ice-covered surface area. Consequently, more open waters allow for higher DMS emissions (Abbatt et al., 2019) and atmospheric oxidation of DMS to MSA involving OH. This can be visualized from the strong coupling between DMS concentration and chlorophyll-a from DMS producing phytoplankton (Park et al., 2013). Moreover, Becagli et al. (2016) concluded that oceanic primary production was related to melting of sea ice and extension of marginal sea ice areas based on satellite derived chlorophyll-a and measurements of MSA (Becagli et al., 2016). Also open leads and marginal ice zones provide primary marine aerosols (Willis et al., 2018). Indeed, previous findings suggest that biogenic productivity in open oceans and sea ice zones and the emission of DMS are responsible for increased new particle formation, as sea ice pack extent retreats (Dall'Osto et al., 2017). Quinn and co-workers reported increased concentrations of MSA at Barrow from 2000 to 2009 associated with the northward migration of the marginal ice zone (Quinn et al., 2009; Sharma et al., 2012; Laing et al., 2013). Of the four northernmost year-round manned observatories at Alert, Mount Zeppelin, VRS and Barrow, the highest MSA concentrations are measured at Mount Zeppelin, likely due to its proximity to open waters around Svalbard, which are a significant source of DMS from May to August (e.g. Lana et al. (2011)). This contrasts with the situation around VRS, which is ice covered most of the year.

Considering the stronger oxidizing environment starting in April, we expect MOA to be abundant until autumn (Chang et al., 2011). MOA constitutes 22% of OA on average during our measurement period ranging from 2-3% of OA in February and March to 24% and 74% of OA in April and May, respectively (Figure 2b and 4). Thus, MOA is by far the most abundant OA from end of April and onwards. MOA dominates the OA mass after polar sunrise and persists during polar daytime so the aerosol's optical impact might be substantial. At the same time, MOA dominates when the overall $PM_1$ concentration is very low, particle numbers are low and hence CCN concentrations can be low. The observed transition between AOA and MOA is in agreement with Narukawa et al. (2008), who observed a transition between fossil fuel influenced OA to marine OA. MOA may contain oxidation products of DMS and other VOCs from oceanic origin, as well as a variety of primary components including sacharides such as mannitol in addition to insoluble gels (Croft et al., 2018; Leck and Bigg, 2005; Orellana et al., 2011; Fu et al., 2013; Ovadnevaite et al., 2011). In line with our findings, modelling at several sites in the Canadian Arctic suggested that marine OA other than MSA may account for more than half of the summertime OA (Croft et al., 2018). These findings encourage further studies of optical properties and chemical composition and physico-chemical parameters as CCN ability or hygroscopicity of aerosols prevailing during polar daytime.

## 4 Conclusion

In the transition from polar night to polar day we observed elevated $PM_1$ concentration ranging from an average of 2.3, 2.3 and 3.3 $\mu g\ m^{-3}$ in February, March and April to 1.2 $\mu g\ m^{-3}$ in May. We concluded $SO_4^{2-}$ to be the most abundant species in sub-micrometer aerosols with highest concentration until the end of April and decreasing in May. This is in accordance with previous findings from VRS, Alert

(Norman et al., 1999) and Svalbard (Udisti et al., 2016) where $SO_4^{2-}$ has been apportioned to be 65% and 75% anthropogenic, respectively. While not previously quantified at VRS, OA was found to be the second largest contributor to $PM_1$ (24%). As for the other species, OA showed a decrease in concentration from mid-April relative to February and March. rBC concentration were found to be highest in the first month and then decreased throughout the campaign – average concentration of 0.2, 0.1, 0.1 and 0.02 µg m$^{-3}$ in February, March, April and May, respectively.

Source apportionment analysis yielded three factors, identified as a Hydrocarbon-like Organic Aerosol (HOA), Arctic haze Organic Aerosol (AOA) and Marine Organic Aerosol (MOA) with O/C ratios of 0.11, 0.63 and 0.95, respectively. HOA, being the least oxidized factor, made up 12% of OA of which 1% of OA was demonstrated to be contamination from the nearby military camp. AOA and MOA made up 86% of OA averaged across the campaign, with AOA averaging 64% and MOA 22% (2% residuals). AOA and MOA showed evidence of SOA. Furthermore, the resemblance of MOA with a previously published marine organic plume where indicative of MOA having a primary organic component. The sum of long-range transported HOA and AOA make-up the vast majority of OA during the Arctic haze period. AOA and MOA exhibit distinct temporal variability. The less oxidized AOA builds up during the Arctic haze period and dominates until early spring (64%-81% of OA), during which both the absolute and relative contribution to the OA burden decreases substantially. In contrast, MOA emerges only after early spring but is then by far the dominating OA from the end of April and onwards (24-74% of OA). The fact that MOA emerges at a time where long-range transport is impaired by increased deposition and a contracting polar dome indicates that the sources to this factor are more Arctic regional in nature. This is supported by the confined atmospheric circulation within the Arctic region during summer (Heidam et al., 2004). This demonstrates the importance of biogenic sources in the Arctic, especially in the spring. In view of changing biogenic processes and corresponding source strengths of aerosol precursors in a changing Arctic climate with changing sea-ice extent, additional high time resolution measurements are urgently needed in order to elucidate the organic components dominating aerosol summer mass and number concentrations.

**Supporting information**

Supporting information describes site information, supplementary instruments, collection efficiency, validation of SP-AMS data, and key diagnostics for the PMF solution.

**Author contribution**

Ingeborg E. Nielsen and Jacob K. Nøjgaard carried out the field measurements, and Ingeborg did the analysis of the SP-AMS data. Jacob and Ingeborg carried out the PMF analysis and took lead in writing the manuscript. Henrik Skov supervised the project and provided critical feedback, participated in the field campaign and helped shape the research and manuscript. Heikki Junninen and Nina Sarnela helped monitor the SP-AMS during the field campaign and commented on the manuscript. Sonya Collier, Qi Zhang and Christopher D. Cappa helped interpret the SP-AMS data set and provided critical feedback

on the manuscript. Andreas Massling and Robert Lange participated in the field campaign and discussed the analysis and commented on the manuscript. Axel C. Eriksson and Manuel Dall'Osto discussed the analysis and results and commented on the manuscript.

**Acknowledgements**

The research was financially supported by the Arctic Centre of Research and iCLIMATE at Aarhus University, Cryosphere-Atmosphere Interaction in a Changing Arctic Climate (CRAICC), WOOD combustion - detailed Monitoring related to Acute effects in Denmark (WOODMAD), the Danish Environmental Protection Agency and Danish Energy Agency with means from the MIKA/DANCEA funds for environmental support to the Arctic Region and the Danish Council for Independent Research (project NUMEN, DFF-FTP-4005-00485B). Special thanks go to laboratory technician Bjarne Jensen and the staff at Station Nord for great support during the field campaign. Sissel Bjørn Svendsen is greatly acknowledged for her data control of the SMPS data and the Villum Foundation is acknowledged for financing the new research station, Villum Research Station, at Station Nord.

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

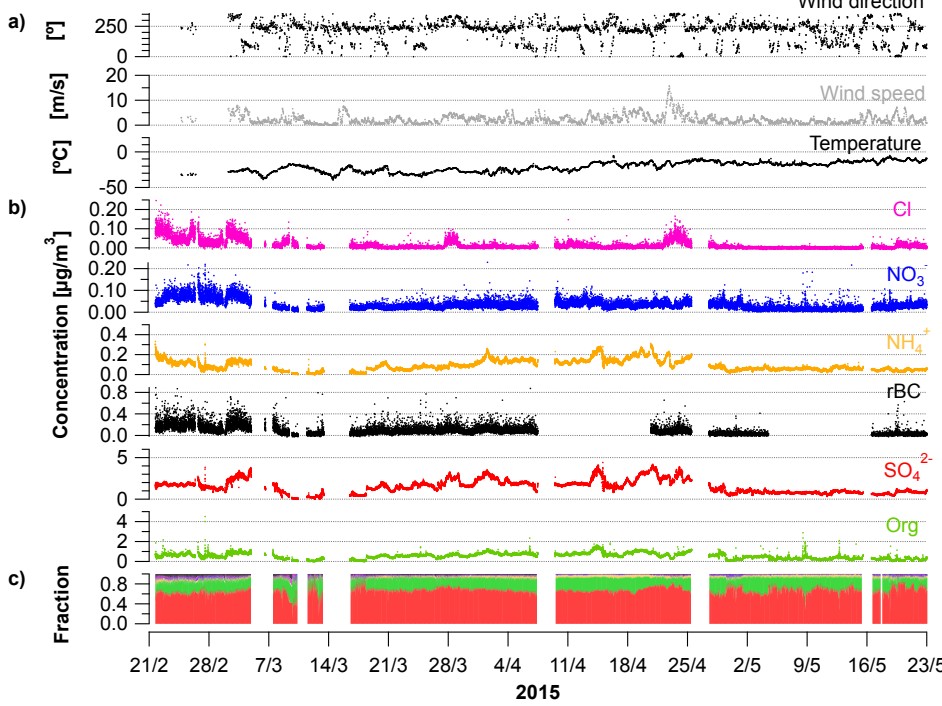

**Figure 1** Time series from 21 February to 23 May 2015 showing a) wind direction [°], mean wind speed [m/s] and temperature [°C], b) concentrations of Cl, $NO_3^-$, $NH_4^+$, rBC, $SO_4^{2-}$ and OA from the SP-AMS [μg/m³], and c) fraction of the aerosol species to the total $PM_1$.

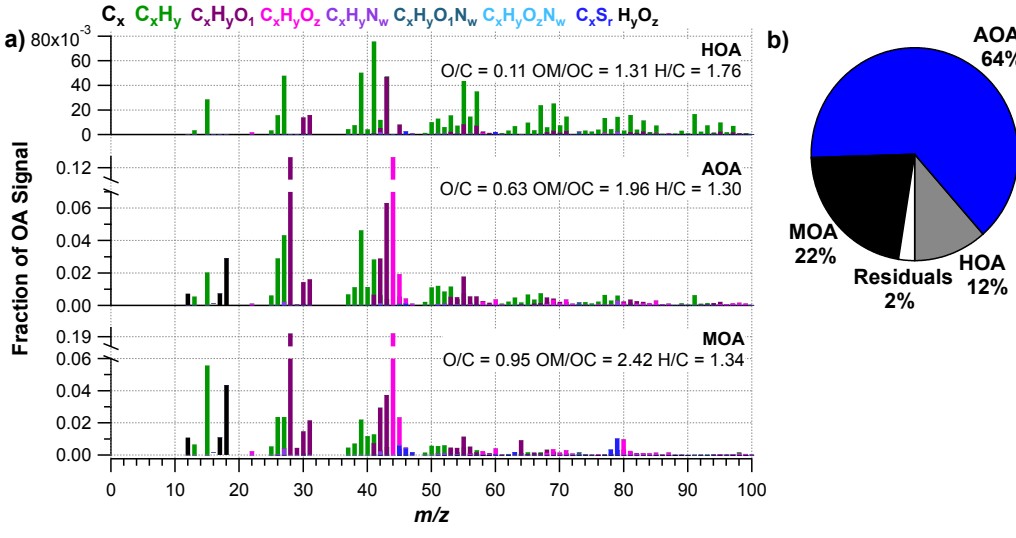

**Figure 2** a) High-resolution mass spectra of PMF factors hydrocarbon-like organic aerosol (HOA), Arctic haze organic aerosol (AOA) and marine organic aerosol (MOA), and b) factor share of ambient mass concentration. O/C, OM/OC and H/C ratio are presented for each factor.

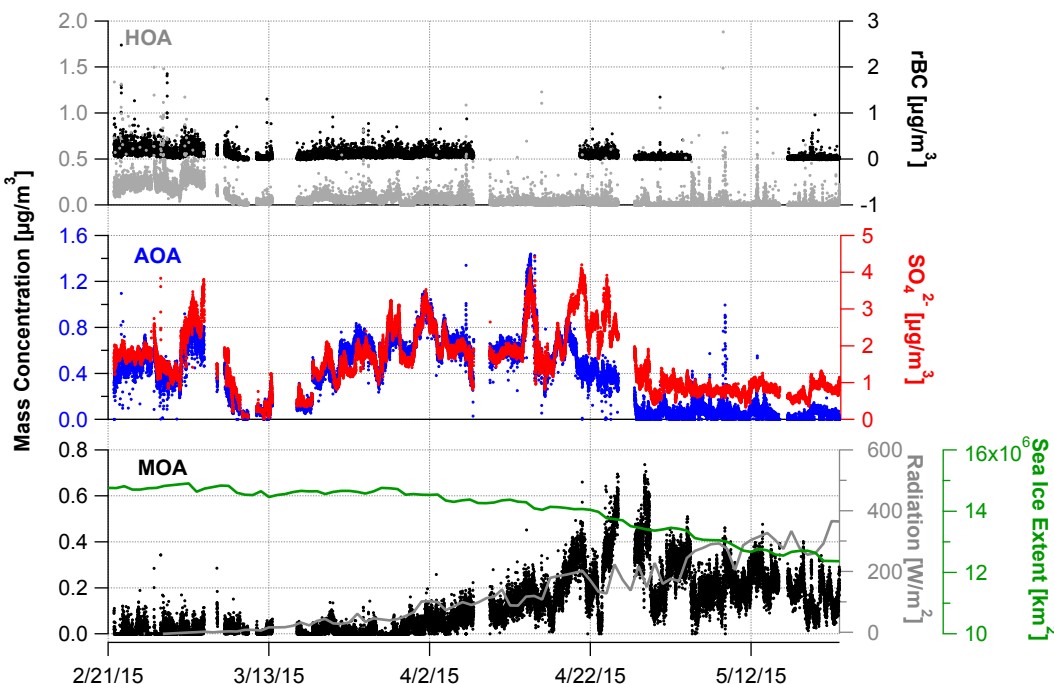

**Figure 3** Time series for hydrocarbon-like organic aerosol (HOA), Arctic haze organic aerosol (AOA), marine organic aerosol (MOA) and tracers (rBC, $SO_4^{2-}$). Sea ice extension on the Northern hemisphere and short-wave radiation (daily average) are included in the time series for MOA (see text).

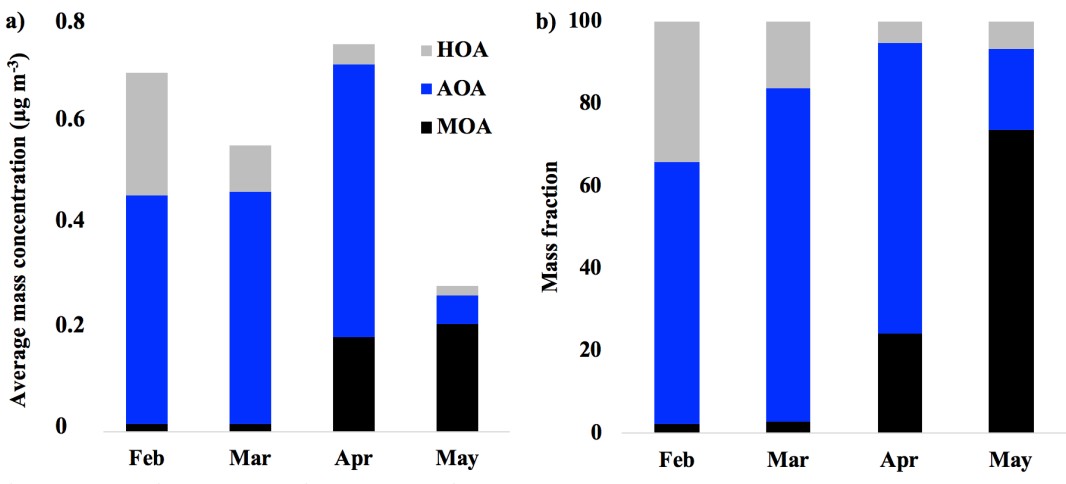

**Figure 4** a) average mass concentration ($\mu g\ m^{-3}$) of hydrocarbon-like organic aerosol (HOA), Arctic haze organic aerosol (AOA) and marine organic aerosol (MOA) in February, March, April and May. b) mass fraction of HOA, AOA and MOA in February, March, April and May.

**Table 1** Detection limits. The detection limits for the SP-AMS is calculated from periods sampling through HEPA filters with a time resolution of 2 minutes (average from eight hepafilter periods of 30 to 60 minutes over the entire campaign). The detection limit for the MAAP is from Massling et al. (2015).

| Instruments | Species | Lower Detection Limit |
|---|---|---|
| **AMS** | HR Org | 0.131 µg m$^{-3}$ |
| | HR SO$_4^{2-}$ | 0.024 µg m$^{-3}$ |
| | HR NO$_3^-$ | 0.021 µg m$^{-3}$ |
| | HR NH$_4^+$ | 0.007 µg m$^{-3}$ |
| | HR Cl | 0.014 µg m$^{-3}$ |
| | HR rBC | 0.010 µg m$^{-3}$ |
| **MAAP** | BC | < 0.006 µg m$^{-3}$ |

1065

**Table 2** R$^2$ correlations between PMF factors and tracers (rBC, MSA, SO$_4^{2-}$ and NH$_4^+$).

| | **HOA** | **AOA** | **MOA** | **rBC** | **MSA** | **SO$_4^{2-}$** | **NH$_4^+$** |
|---|---|---|---|---|---|---|---|
| **HOA** | - | 0.08 | 0.11 | 0.35 | 0.13 | 0.08 | 0.04 |
| **AOA** | - | - | 0.14 | 0.21 | 0.27 | **0.67** | 0.49 |
| **MOA** | - | - | - | 0.07 | 0.68 | 0.00 | 0.03 |
| **rBC** | - | - | - | - | 0.08 | 0.18 | 0.15 |
| **MSA** | - | - | - | - | - | 0.02 | 0.00 |
| **SO$_4^{2-}$** | - | - | - | - | - | - | 0.70 |
| **NH$_4^+$** | - | - | - | - | - | - | - |