# Peer review of "Biogenic and anthropogenic sources of aerosols at the high Arctic site Villum Research Station"

_Atmospheric Chemistry and Physics, 2019_

## Referee Comment (RC1) · Anonymous Referee #1 · 20 Apr 2019

This paper represents a significant contribution to furthering our understanding of Arctic aerosol, and how its chemistry evolves over the transition from dark winter to light spring. This work reports on three months (February - May) of high time resolution measurements of PM1 aerosol composition at Villum Research Station, at Station Nord, Greenland. In particular, this paper provides some of the most chemically detailed measurements available of the evolution of organic aerosol chemistry in the High Arctic during the transition between winter and spring. These results are in line with, and add to, our current understanding of Arctic organic aerosol. In the specific comments I have made some suggestions for additional data analysis, and while these suggestions may improve the paper I do not consider them required for publication. The comments and suggestions below are meant to help further improve an already

very interesting and thorough paper.

General Comments: (1) The large amount of data and time span of these measurements from winter to spring mean that a number of interesting conclusions and observations are presented in this paper. The paper may benefit from sharpening the focus on what the authors feel are the most valuable or interesting conclusions. The main findings are somewhat obscured in the abstract and conclusions in the current version of the paper. For example, the abstract begins by highlighting the importance of BC aerosol in the Arctic, but by the end of the conclusions it is fairly clear that results related to the evolution of organic aerosol may be a larger outcome of this work. I suggest the authors revise some information in the abstract and conclusions to best highlight the largest conclusions of this work.

(2) The title is currently very general, and does not highlight the main findings of the work. I suggest the authors revise toward a more declarative title that helps highlight their specific findings.

(3) Throughout the paper the text appears to suggest that secondary organic aerosol and marine emissions are mutually exclusive (e.g., L37-38). If by marine emissions the authors mean only sea spray, then I suggest they state this clearly. Marine organic aerosol can arise from both primary and secondary processes, and it is extremely difficult to distinguish unambiguously between primary and secondary marine aerosol. With the evidence presented in this paper, the authors cannot reliably determine if the marine organic aerosol factor they measure at VRS is primary or secondary. I suggest that the authors can acknowledge these challenges while still highlighting the evidence that they do have for each mechanism (e.g., the presence of MSA in the MOA factor showing that secondary chemistry contributes to MOA).

Specific Comments:

L22-27 (and elsewhere): Given that the authors present novel measurements of aerosol chemistry over the period of transition from a dominance of long range transported Arctic Haze to a cleaner regime more dominated by inner-Arctic sources, it may not be particularly useful to quote average concentrations and mass fractions over the whole study period. One perspective is that the most important aspect of these measurements is that they cover this period of transition from anthropogenic to biogenic sources. Further, these mean values are likely dominated by the larger amount of data covering the Arctic haze period, and so are more reflective of the average composition in only on regime of the Arctic atmosphere.

L26: "Arctic Haze leveling-off" may not reflect our current understanding of how source regions and removal changes over this period

L63-64: While it is true that distinct layers of aerosol are often associated with Arctic Haze, it is also true that elevated levels of aerosol pervade through the depth of the troposphere, at least within the polar dome, during this time.

L89: What "this" is could be clarified

L94-95: Another reference that could be included here: Leaitch et al., ACP, 2018 https://doi.org/10.5194/acp-18-3269-2018. The author's work is an extremely useful follow on from the lower time resolution work on OA at Alert.

L103-104: Revise "marine aerosols is a source of inorganic and organic aerosols" for clarity

L105: Are the "other mechanisms" worth elaborating here?

L108: Other work about sea salt in Arctic regions (e.g., Huang 2017 https://www.atmos-chem-phys.net/17/3699/2017/) may be worth including here

L188-189: The NH4 RIE can often differ significantly from 4, what value do the authors obtain when they calculate NH4 RIE from their NH4NO3 calibrations?

L191-192: While I agree that the Cl family of peaks likely comes from non-refractory chloride, Ovadnevaite et al (https://doi.org/10.1029/2011JD017379) have shown that

the signal for NaCl+ can be used as an indication for the presence of sea salt. Do the authors observe significant NaCl+ in their data set?

L197-199: Why not constrain C1+ to the expected ratio from Regal Black (or other material) and fit C2+ through C5+?

L229-232: The apparent RIE for rBC is a convolution of the true RIE and collection efficiency. Also, uncertainty in the MAC applied to the MAAP data could contribute to this discrepancy. While I don't dispute the choice the authors have made in scaling SP-AMS rBC data to the MAAP, we know that all these approaches to measuring BC carry uncertainty and the MAAP may not provide the most accurate measurement in the case of an aged, highly internally mixed black carbon containing aerosol. If available, a direct comparison between the SP-AMS and the MAAP during calibration with representative BC materials may be useful here.

L245: How much of an effect does scaling SP-AMS rBC to the MAAP have on the comparison between AMS total PM1 and SMPS PM1?

L247-250: During the beginning few days of the study the SMPS concentration is higher than the SP-AMS concentration, and this is the only period where this appears to be the case. SP-AMS chloride was elevated during this time; was any measurable signal for NaCl+ present at that time?

L291-293: It may be worth clarifying this. Long range transport suggests to me sources far outside the Arctic, but the authors suggest that this is likely not the case after the transition to cleaner conditions. Or, do the authors suspect that the MOA was transported from more southerly marine regions? It may be useful to provide some general indication of the meteorological regime or air mass histories, for example, for before and after the Arctic Haze decreases substantially at Villum.

L300-301: Long range transport from marine regions can mean that a portion of springtime Arctic sulphate is from DMS oxidation (https://www.atmos-chemphys.net/17/8757/2017/)

L313: Is the contribution of MSA subtracted from reported organics and sulphate?

L315: Is this a statement based only on the measurement period presented in this study. Measurements of MSA at Alert would suggest that DMS peaks later in the summer (Leaitch et al., Elementa, 2013: https://www.elementascience.org/articles/10.12952/journal.elementa.000017/)

L331: Biomass burning may be a larger source to the Arctic than farms

L342: Current evidence suggests that frost flowers may not be an important source of aerosol (see for example, Huang 2017 https://www.atmos-chem-phys.net/17/3699/2017/ and references within)

L351-353: This type of information may belong more in the introduction, rather than the results and discussion

L364-366: The authors need to screen the data based on wind direction, or another measured parameter, before reporting and interpreting R2 values here.

L385-395: A mean pToF size distribution for the Arctic Haze period and the more biogenically influenced period could help with this interpretation of mixing state (and would be very interesting!).

L406: Secondary or highly aged primary organic aerosol, it is difficult to interpret a mechanism based solely on CO2+ contribution alone. I suggest the authors elaborate on how they reached this conclusion

L412: It might be useful to indicate the significance of these correlations in Table 2 in some way (e.g., bold R2 values)

L441-442: It is really difficult to attribute primary or secondary sources from the mass spectrum alone. Marine OA observed at Mace Head is likely a combination of primary and secondary OA. Ovadnevaite et al., GRL 2011

(https://doi.org/10.1029/2010GL046083) state (paragraph 9): "The relatively high amount of oxygenated organics typically indicates a chemical aging of the aerosol [Jimenez et al., 2009] with possible contributions from both oxidation of primary aerosol organics and SOA (secondary organic aerosol) formation."

L 454-455: "In April, the highest OA concentrations is observed where AOA accounts for around 70% of OA (Figure 4). In May, MOA becomes the dominating OA while AOA nearly disappears." This seems to be an important point of the paper, which could be further highlighted in the abstract/conclusions and results & discussion sections.

L456-457: While comparing to this Alert study is valuable, Narukawa use a very different method and their data represent measurements from 15 years prior to this study. A direct comparison is difficult to make, but I agree it is interesting despite these differences.

L461-462: Some additional references related to DMS would be useful here. The marginal ice zone is also important for DMS

L465-471: Some more detailed information about the source regions impacting Villum during winter and spring might help this discussion and interpretation.

L473-474: Why speculate about the emergence of a continental factor?

L474: Reporting an overall average MOA fraction here is a bit confusing, since the previous discussion demonstrates its much higher contribution once the AOA decreases.

L475-477: In addition, and perhaps more importantly MOA dominates the organic aerosol mass when the overall concentrations are very low, particle numbers are low, and so cloud condensation nuclei concentrations can be low.

L480-481: Oxidation of DMS and other VOCs would be considered secondary. The wording of this sentence is a bit confusing

L482-483: In Croft ACP 2019, secondary OA accounted for up to half of the summertime OA, and primary marine OA also contributed. The authors may want to be more clear in their usage of marine OA, primary OA and secondary OA. Marine OA can come from both primary and secondary processes.

L484-487: This introductory information may fit better within the introduction.

L493-495: Comparing to Alert may also be warranted, given its proximity to VRS

L495-500: The authors' clear observations of changing OA character and sources over the winter to spring to late spring transition may be a more important conclusion that that these organic species can be mixed with rBC.

L506-507: The observations presented here cannot unambiguously determine whether AOA and MOA is primary or secondary in origin. The mass spectrum similar to Ovadnevaite 2011, only suggests that the aerosol is marine in origin. More information would be needed to suggest a dominant formation process. While the correlation of AOA with sulphate may suggest secondary processes, this aerosol is also transported over very long distances and so aerosol from somewhat different formation processes may co-vary in time at such a remote location.

L512-514: I agree in general with this statement, but some more information about source regions impacting Villum would go a long way in this interpretation. Further, do the authors have access to CO data that could potentially help to demonstrate the increase in deposition mentioned here? (e.g., see Garrett et al., GRL, 2011 doi:10.1029/2011GL048221)

Figure 3: That the authors observe a distinct HOA factor in Arctic haze that co-varies in time more closely with rBC than with AOA or sulphate is interesting. Intuitively I would expect Arctic haze aerosol to be overall extremely oxidized, though the prevalence of HOA in the dark winter suggests not. Do the authors have specific evidence to show that the HOA was not more regionally sourced than the AOA? Do polar plots of wind direction/speed and PNF factor intensity shed any light on differences in source

regions?

Figure 4: Does the MSA-to-sulphate ratio, and organic-to-sulphate ratio, increase in a similar manner to MOA on a monthly basis?
* * *

---

## Referee Comment (RC2) · Anonymous Referee #3 · 22 Apr 2019

The paper presents an analysis of aerosol composition measured at the Villum Research Station in Northern Greenland based on SP-AMS data collected over 3 months in 2015. There is a lack of measurements of organic aerosol in the Arctic and, in particular, measurements of the composition of the organics and how that changes with season. Hence, the data provided here make a substantial contribution to our understanding of the importance of marine and anthropogenic sources of organics to the Arctic atmosphere with changes in solar radiation levels and sea ice extent. The paper is well-written and the balance between material in the main text and the SI is appropriate. I only a few minor concerns which are listed below.

Lines 24 – 25: Do "organic matter" and "organic aerosol" both refer to organic aerosol concentrations as ug C/m3 or as total particulate organic matter including H and O?

[Figure]

Lines 78 – 79: Decreasing trends in nss SO4 and BC have been documented for Barrow. Please see Chapter 9 of the 2015 AMAP report on Black Carbon and ozone as Arctic climate forcers (www.amap.no).

Line 214: Applying a uniform specific absorption coefficient for BC could affect temporal variability if the nature of the BC (source, aging processes, etc.) lead to varying specific absorption coefficients.

Lines 248 – 249 and SI lines 85 – 98: It is not clear from the main text that periods where differences between PM1 determined from the SP-AMS and the SMPS were at least 2 ug/m3 (late March/early April and mid-April) were excluded from the data analysis. It states in the SI that data from Feb 21 – 26 and Mar 29 – Apr 2 were excluded. Please clarify this in the main text. Also – what is the impact of not including sea salt in the SP-AMS derived PM1 since it will be included in the SMPS PM1? The modal number diameter of the sea salt mode is ∼200 to 300 nm so should be detected by the SMPS.

Lines 312 - 315: What is the MSA to SO4 ratio during periods when MSA was detected? Can the ratio be used to assess the importance of biogenic vs. anthropogenic sources of SO4?

Line 340 – 342: Is the attribution of Cl and NO3 to frost flowers (i.e., a local source) due to their presence in the supermicron size range? Please clarify in the main text.

---

## Referee Comment (RC3) · Anonymous Referee #2 · 26 Apr 2019

The study reports on SP-HR-AMS measurements conducted at Villum Research Station in the north of Greenland from February to May 2015. The authors investigate the concentrations and evolution of refractory black carbon (rBC), particulate sulfate (SO4) and organic aerosol (OA). The first half of the manuscript focuses on rBC, the second on OA that was further investigated by conducting positive matrix factorization (PMF). Three factors were identified: hydrocarbon-like OA (HOA) with the smallest contribution, Arctic haze OA (AOA) with the largest contribution and marine OA (MOA).

Detailed measurements of rBC and OA in the high Arctic are rare, especially outside of the summer season. The real strength of this study are the real-time observations during the transition period from winter to spring when sunlight returns and Arctic haze conditions fade. While the authors make this point, they also "dilute" their message

by putting emphasis on reporting average concentrations for the entire study period, which do not address the environmental change. Generally, this study provides valuable insights into the aerosol chemical composition in the high Arctic and should be published with major revisions as suggested below. General and specific comments are mentioned below, all other comments are highlighted in the attachment.

General comments:

A shortcoming of the study is that it underexplores the HR-AMS data. There is no reporting of hetero-atoms such as nitrogen or sulfur in the OA. The contribution of those as a function of time could reveal more details about the sources of MOA in particular. At the moment only O:C ratios are provided. I suggest exploring also the N- and S-containing contributions to OA. In particular the contribution of MSA should be quantified. MSA is discussed in the manuscript (l. 437ff), but rather superficially. See also respective comment in the manuscript.

The authors mention often the average concentrations of the constituents during the campaign. As mentioned above the real strength of the observations lies in having captured the transition periods and the transition cannot be described by campaign average but should rather be discussed as gradients are differences. How long does the transition take, which markers change first, which ones later, or all simultaneously? I suggest changing the emphasis to transition characterization throughout the whole manuscript. For example: l. 345: here an average BC concentration is mentioned; l. 367: a slope or gradient for the SO4 concentration would make more sense here;

I suggest renaming the title to "Biogenic and Anthropogenic sources of Arctic Aerosols at Villum Research Station". "Arctic Aerosols" alone is misleading, because the measurements reflect the unique environment of VRS in northern Greenland. That is very different from the Canadian archipelago or Svalbard as the authors write themselves. Along the same line is the inaccuracy with which the authors cite literature in the introduction:

[Figure]

- L. 37: How do the authors define the "Arctic summer aerosols"? Do they mean the high Arctic, so basically the Arctic Ocean? Or do they include terrestrial parts of the Arctic. This makes a fundamental difference for the composition and other properties of aerosols.

- L. 79: This information is incomplete. The paper also states that SO4 decreased significantly in Alert and Zeppelin and that the lack of a trend at Barrow is likely due to the limited data coverage. This information needs to be added.

- L. 86: This article is focused on the Canadian Arctic mostly. Use literature that is more relevant to the entire Arctic. Furthermore, the article has been published in 2019 in ACP.

- L. 112 "DMS emissions in the Arctic have increased by 30 %..." Is this true for the entire Arctic or the Canadian sector? It is important to provide a differentiated picture of what is happening, otherwise false impressions are created.

- L. 114: "demonstrated" is an overstatement, the paper infers. The authors show the relationship but do not provide an explanation.

- L. 115: MSA does not nucleate or form new particles, it rather condenses and grows particles.

- L. 116: It is not only believed that ammonia comes from sea bird colonies, this has been shown multiple times. There are global inventories for ammonia seabird emissions even.

Specific comments:

L. 23: unclear whether the particulate sulfate or PM1 amounted to 2.3 ug / m3

l. 40: Why is it urgently needed to elucidate the chemical components? The authors probably mean that modeling the future of the Arctic requires process understanding. Just because climate is changing doesn't mean we need highly time resolved aerosol

data.

l. 45: consider referring to the special IPCC report on 1.5 °C and the AMAP 2015 report on BC and ozone in the Arctic.

l. 52: ice does not condense onto particles

l. 63: Is it truly "visible"? Strong haze events might be visible by eye, but the typical Arctic Haze is still orders of magnitude lower in mass concentrations as the visible urban air pollution, as is somehow inferred by this sentence.

l. 67: As it is written it contradicts above statement that says that Arctic Haze sources are located within in the polar dome. This needs some clarification or more exact formulation.

l. 87: why should vegetation fires not be considerable? It's a question of whether their emissions are transported to the high Arctic.

l. 93: Consider referring also to Chang et al., 2011, ACP doi:10.5194/acp-11-10619-2011They characterize PM1 aerosol measured with an AMS and PMF in the central Arctic during the ASCOS campaign. Also Willis et al., 2018, 10.1029/2018RG000602 provide and overview of what we know about Arctic aerosol and it's detailed composition.

l. 98 ff: This seems to be more a concluding statement which should be placed later. It is a bit awkward after the OA discussion.

l. 108: the explanation why the role is important is missing.

l. 110: Unclear where MSA is increasing.

l. 123: revise the sentence, it is grammatically incorrect and does not list the two disadvantages.

l. 126: delete "and trends". Trends are longer term changes.

l. 139: PMF cannot reveal source regions just source types.

l. 153: Where is the HVS data used? This is not evident in the manuscript. If they are used that needs to be stated and then more information like flowrate, sample duration etc. needs to be added, or a references to the supplement needs to be given.

l. 176: "inspected" "inspected" sounds like the flow rate was measured once. I hope it was checked several times during the campaign.

l. 176 if the size calibration was conducted with ammonium nitrate, a DMA must have been operated as well to select a range of sizes. This information is missing entirely.

l. 179: Why was there no determination of the relative ionization efficiency of sulfate with ammonium sulfate?

l. 191: The AMS also sees NaCl, see Ovadnevaite et al., 2012, doi:10.1029/2011JD017379. and other publications. The influence of NaCl needs to be considered as well.

l. 214: add manufacturer and model number of the SMPS.

l. 224: "majority". Can the authors be more specific and provide the quantiles?

l. 253: the sentence is confusing.

l. 273: "chemical composition" instead of "chemistry"

l. 285: A comparison to other studies is missing that would reveal why the concentration can be perceived as relatively high.

l. 302: What is the role of light here?

l. 303: "at its source region" This should rather read: "in the vicinity of the source region, " SO2 oxidation does not happen immediately and normally SO2 has already been transported away some distance from the source before it is oxidized to SO4 2-

l. 305: Figure 3 is mentioned before Figure 2.

l. 308: "originating from Siberia" Is this not a contradiction to the main wind direction from the south-west? How representative is the wind direction of the general atmospheric circulation around VRS?

l. 319: How do you define spring season? In my understanding mid-April and later is spring. So the sentence does not make sense to me.

l. 323: Would the pollution from the military not result in a separate PMF factor? Or is the HOA that is long-range transported so similar to the fresh HOA?

l. 331: Is this also true for winter? Are there birds all year around?

l. 335: Add a reference for the longer lifetime.

l. 336: Please correct Cl to Cl- throughout the manuscript.

l. 339: should be chloride and not chlorine

l. 361: Is this true that the sources are the same for the entire Arctic, for all seasons or the Haze period where you have long lifetimes and hence rather well mixed conditions?

l. 364 – 366: To me it doesn't make sense to include local contamination periods for a general conclusion on rBC and SO4 correlation. I suggest removing the local influence first and then redoing the correlation analysis.

l. 407: "AOA is abundant during February to mid-April..." this is redundant. The sentences before that say the same.

l. 421: I cannot follow the argument. What is the contribution quantitatively and what would be expected from the literature? Is the literature appropriate for a comparison?

l. 433: Please be more specific in how far it resembles the Mace Head spectrum.

l. 443: How does the MOA factor resemble HR-AMS spectra from the Southern Ocean? doi:10.5194/acp-13-8669-2013 Can the authors discuss whether the MOA factor is more universal, i.e. VRS, Mace Head, other oceans?

[Figure]

l. 456:What is the lowest concentration of OA?

l. 456: What does "this" refer to? The concentration of OA or the 75 % MOA in the OA?

l. 475ff: This sentence is confusing. I do not understand the main message.

l. 480: "oxidation products of DMS and other VOCs" These are also secondary. The argument does not make sense like this.

l. 481: Âńprimary components including colloidal gels…Âż As far as I read the sentence MOA is the specific factor found by the authors using the HR-AMS. So the question is whether the primary compounds like gels would actually be seen in the MOA factor? To my knowledge they evaporate at temperatures higher than 600 °C. This means that generally marine organic aerosol can contain these compounds, but the MOA factor likely doesn't due to instrumental limitations.

l. 487: enhancement through the lensing effect?

l. 494f: 75 + 3 + 12 + 12 is > 100 %.

l. 503 What does "reduced" mean? The least amount of oxygen?

Figure 2: I suggest to either make the axis logarithmic or but them off at 0.05 (with indicating the true extent of the big peaks) to make the pattern visible. The AOA and MOA spectra are not informative like they are now because on cannot see anything.

Figure 3: I suggest to move the rBC trace up. It's not visible like this and hence not usefull.

Figures S3: the figures have very low resolution.

Figure S2: The y-axis could start at 0.5.

Please also note the supplement to this comment:
https://www.atmos-chem-phys-discuss.net/acp-2019-130/acp-2019-130-RC3-

supplement.pdf

**Supplement:**

[revised manuscript text omitted]

---

## Author Comment (AC1) · 12 Jul 2019

**Response to Referees**
**Referee #1**

**This paper represents a significant contribution to furthering our understanding of Arctic aerosol, and how its chemistry evolves over the transition from dark winter to light spring. This work reports on three months (February - May) of high time resolution measurements of PM1 aerosol composition at Villum Research Station, at Station Nord, Greenland. In particular, this paper provides some of the most chemically detailed measurements available of the evolution of organic aerosol chemistry in the High Arctic during the transition between winter and spring. These results are in line with, and add to, our current understanding of Arctic organic aerosol. In the specific comments I have made some suggestions for additional data analysis, and while these suggestions may improve the paper I do not consider them required for publication. The comments and suggestions below are meant to help further improve an already very interesting and thorough paper.**

**General Comments:**

1. **The large amount of data and time span of these measurements from winter to spring mean that a number of interesting conclusions and observations are presented in this paper. The paper may benefit from sharpening the focus on what the authors feel are the most valuable or interesting conclusions. The main findings are somewhat obscured in the abstract and conclusions in the current version of the paper. For example, the abstract begins by highlighting the importance of BC aerosol in the Arctic, but by the end of the conclusions it is fairly clear that results related to the evolution of organic aerosol may be a larger outcome of this work. I suggest the authors revise some information in the abstract and conclusions to best highlight the largest conclusions of this work.**
We agree with the Referee and has rewritten line 20-21: *"There are limited measurements of the chemical composition, abundance, and sources of atmospheric particles in the high Arctic."*
In the Conclusion section, we have deleted the passage: *"OA and $SO_4^{2-}$ have the potential to condense on and coat black carbon, potentially impacting the CCN activity of and light absorption by BC. However, the chemical composition should be further studied in summer and autumn."*

2. **The title is currently very general, and does not highlight the main findings of the work. I suggest the authors revise toward a more declarative title that helps highlight their specific findings.**
We thank the Referee for this valuable comment, which was also suggested by Referee no. 2. To accommodate both Referees we have changed the title to: *"Biogenic and anthropogenic sources of aerosols at the high Arctic site Villum Research Station"*.

3. **Throughout the paper the text appears to suggest that secondary organic aerosol and marine emissions are mutually exclusive (e.g., L37-38). If by marine emissions the authors mean only sea spray, then I suggest they state this clearly. Marine organic aerosol can arise from both primary and secondary processes, and it is extremely difficult to distinguish unambiguously between primary and secondary marine aerosol. With the evidence presented in this paper, the authors cannot reliably determine if the marine organic aerosol factor they measure at VRS is primary or secondary. I suggest that the authors can acknowledge these challenges while still highlighting the evidence that they do have for each mechanism (e.g., the presence of MSA in the MOA factor showing that secondary chemistry contributes to MOA).**
We agree with the Referee and recognize that marine POA and SOA could be perceived as mutually exclusive, though this was not our intension. We have corrected this, including in line 40-42: *"Our data supports current understanding that Arctic aerosols are highly influenced by secondary aerosol formation, and with an important contribution from marine emissions during Arctic spring in remote high Arctic areas."*

**Specific Comments:**

1. **L22-27 (and elsewhere): Given that the authors present novel measurements of aerosol chemistry over the period of transition from a dominance of long range transported Arctic Haze to a cleaner regime more dominated by inner-Arctic sources, it may not be particularly useful to quote average concentrations and mass fractions over the whole study period. One perspective is that the most important aspect of these measurements is that they cover this period of transition from anthropogenic to biogenic sources. Further, these mean values are likely dominated by the larger amount of data covering the Arctic haze period, and so are more reflective of the average composition in only on regime of the Arctic atmosphere.**

This is a very sound and valid comment and we thank the Referee for the suggestion. We have now removed the average concentrations from the abstract and rewritten some of the sentences to accommodate this comment (marked in bold): Line 23-39:

*"**During this period, we observed the Arctic haze phenomenon with elevated $PM_1$ concentration ranging from an average of 2.3, 2.3 and 3.3 $\mu g\ m^{-3}$ in February, March and April to 1.2 $\mu g\ m^{-3}$ in May.** Particulate sulfate ($SO_4^{2-}$) accounted for 66% of the non-refractory $PM_1$ **with highest concentration until the end of April and decreasing in May.** The second most abundant species was organic aerosol (OA) (24%). Both OA and $PM_1$, estimated from the sum of all collected species, showed a marked decrease throughout May in accordance with the polar front moving North together with changes in aerosol removal processes. **The highest refractory black carbon (rBC) concentrations were found in the first month of the campaign averaging 0.2 $\mu g/m^3$. In March and April, rBC averaged 0.1 $\mu g/m^3$ while decreasing to 0.02 $\mu g/m^3$ in May.***

*Positive Matrix Factorization (PMF) of the OA mass spectra yielded three factors: (1) a Hydrocarbon-like Organic Aerosol (HOA) factor, which was dominated by primary aerosols and accounted for 12% of OA mass; (2) an Arctic haze Organic Aerosol (AOA) factor; and (3) a more oxygenated Marine Organic Aerosol (MOA) factor. AOA **dominated until mid-April (64%-81% of OA)**, while being nearly absent from the end of May and correlated significantly with $SO_4^{2-}$, suggesting the main part of that factor being secondary OA. The MOA emerged late at the end of March, where it increased with solar radiation and reduced sea ice extent, and dominated OA for the rest of the campaign until the end of May **(24-74% of OA), while AOA was nearly absent."***

In addition, we have gone through the entire manuscript and corrected/changed paragraphs where average campaign concentrations were presented. Changes are shown below (marked in bold):

- Line 294-296: We added ranges to *"The total measured $PM_1$ concentration during the field study may seem relatively high, averaging 2.3 $\mu g\ m^{-3}$**- ranging from 2.3, 2.3 and 3.3 $\mu g\ m^{-3}$ in February, March and April to 1.2 $\mu g\ m^{-3}$ in May."***
- Line 317-319: We deleted the average $SO_4^{2-}$ concentration and changed it to: *"During the entire campaign, $SO_4^{2-}$ is the dominant species that on average makes up almost 70% of the $PM_1$ mass concentration **with highest concentration until the end of April and decreasing in May (Figure 1b-c)."***
- Line 342-343: We deleted the average OA concentration and changed it to: *"In this study, the OA fraction is the second largest contributor to $PM_1$ **where** weekly averages showed a clear decrease from mid-April relative **to concentrations in February and March** concentrations (Figure 1)."*
- Line 349-350: We deleted the average concentration and changed it to: *"Particulate $NH_4^+$ is found in much lower concentrations compared to OA and $SO_4^{2-}$ **but with the same transition pattern as the two other species."***
- Line 361-362: We have kept the mentioning of the average concentration of $NO_3^-$ and Cl due the comparison with the detection limit of the species.
- Line 374-375: We have deleted the campaign average for rBC and instead looked at concentration for the different months: *"The highest rBC loadings are found in the first month of the campaign (February) averaging 0.2 $\mu g/m^3$. In March and April, the average is 0.1 $\mu g/m^3$ which then decreases to 0.02 $\mu g/m^3$ in May."*

- Line 440-442: We have added more information concerning the AOA development: *"AOA accounts for 64% of OA mass for the entire field study **but ranges from 64%, 81% and 71% of OA in February, March and April to 20% in May (Figure 2b and 4)."***
- Line 517-519: We have added more information concerning the MOA development: *"MOA constitutes 22% of OA on average during our measurement period **ranging from 2-3% of OA in February and March to 24% and 74% of OA in April and May, respectively** (Figure 2b and 4)."*
- Line 550-554: We have added more information concerning the AOA and MOA development in the conclusion: *"The less oxidized AOA builds up during the Arctic haze period and dominates until early spring **(64%-81% of OA),** during which both the absolute and relative contribution to the OA burden decreases substantially. In contrast, the MOA is nearly non-existent until early spring but is then by far the dominating OA from the end of April and onwards **(24-74% of OA)."***

Finally, we have changed the first paragraph of the conclusion to accommodate this comment (marked in bold): Line 532-541:

*"In the transition from polar night to polar **day we observed elevated PM₁ concentration ranging from an average of 2.3, 2.3 and 3.3 µg m-3 in February, March and April to 1.2 µg m-3 in May.** We concluded SO42- to be the most abundant species in sub-micrometer aerosols **with highest concentration until the end of April and decreasing in May.** This is in accordance with previous findings from VRS, Alert (Norman et al., 1999) and Svalbard (Udisti et al., 2016) where SO42- has been apportioned to be 65% and 75% anthropogenic, respectively. **While not previously quantified at VRS, OA was found to be the second largest contributor to PM1 (24%). As for the other species, OA showed a decrease in concentration from mid-April relative to February and March. rBC concentration were found to be highest in the first month and then decreased throughout the campaign – average concentration of 0.2, 0.1, 0.1 and 0.02 µg m-3 in February, March, April and May, respectively."***

2. **L26: "Arctic Haze leveling-off" may not reflect our current understanding of how source regions and removal changes over this period**
   This is correct and we have changed the sentence to: Line 28-29: *"…marked decrease throughout May in accordance with the polar front moving North together with changes in aerosol removal processes."*

3. **L63-64: While it is true that distinct layers of aerosol are often associated with Arctic Haze, it is also true that elevated levels of aerosol pervade through the depth of the troposphere, at least within the polar dome, during this time.**
   We have changed the sentence to: Line 70-71: *"The Arctic haze peaks in early spring (Heidam et al., 1999; Law and Stohl, 2007; Stohl, 2006; Heidam et al., 2004; Abbatt et al., 2019)."*

4. **L89: What "this" is could be clarified**
   To clarify the sentence, we have made the following changes: Line 94-97: *"Overall, the general seasonal cycle of BC in the Arctic is characterized by highest concentrations observed between January and April and lowest concentrations throughout the summer, but with periodic spikes in concentration throughout the summer."*

5. **L94-95: Another reference that could be included here: Leaitch et al., ACP, 2018 https://doi.org/10.5194/acp-18-3269-2018. The author's work is an extremely useful follow on from the lower time resolution work on OA at Alert.**
   Thank you for the suggestion and we have now added the reference: Line 100-102: *"…though few studies have characterized this component in detail (Barrett et al., 2015; Brock et al., 2011; Frossard et al., 2011; Kawamura et al., 2010; Quinn et al., 2002; Shaw et al., 2010; Leaitch et al., 2018; Chang et al., 2011; Willis et al., 2018)."*

6. **L103-104: Revise "marine aerosols is a source of inorganic and organic aerosols" for clarity**
   We have revised the sentence for clarity: Line 107-109: *"Marine and coastal marine locations constitute a large part of Arctic, and marine aerosols comprise both organic and inorganic constituents of primary and secondary origin."*

7. **L105: Are the "other mechanisms" worth elaborating here?**
   We have chosen not to elaborate since we don't think it is within the scope of this work.

8. **L108: Other work about sea salt in Arctic regions (e.g., Huang 2017 https://www.atmos-chem-phys.net/17/3699/2017/) may be worth including here**
   Based on another Referee's comment we have chosen to rewrite the sentence to: Line 112-113: *"Marine aerosols play an important role for the climate due to their optical properties and ability to alter cloud nucleation (Abbatt et al., 2019; Willis et al., 2018)."*

9. **L188-189: The NH4 RIE can often differ significantly from 4, what value do the authors obtain when they calculate NH4 RIE from their NH4NO3 calibrations?**
   We thank the Referee for addressing this and we have now determined the average $RIE_{NH4}$ to 3.5. The relevant figures and numbers have been changed accordingly and the following sentence has been added: Line 97-98: *"A RIE of 3.5 was applied for $NH_4^+$."*

10. **L191-192: While I agree that the Cl family of peaks likely comes from non-refractory chloride, Ovadnevaite et al (https://doi.org/10.1029/2011JD017379) have shown that the signal for NaCl+ can be used as an indication for the presence of sea salt. Do the authors observe significant NaCl+ in their data set?**
    This is an interesting point but at the same time it is beyond the scope of this manuscript.

11. **L197-199: Why not constrain C1+ to the expected ratio from Regal Black (or other material) and fit C2+ through C5+?**
    We have used this method previously with success and have therefore found it to be a good solution for this manuscript as well – see Nielsen et al., 2017 (doi.org/10.1016/j.atmosenv.2017.06.033) and Martinsson et al., 2015 (doi.org/10.1021/acs.est.5b03205).

12. **L229-232: The apparent RIE for rBC is a convolution of the true RIE and collection efficiency. Also, uncertainty in the MAC applied to the MAAP data could contribute to this discrepancy. While I don't dispute the choice the authors have made in scaling SP-AMS rBC data to the MAAP, we know that all these approaches to measuring BC carry uncertainty and the MAAP may not provide the most accurate measurement in the case of an aged, highly internally mixed black carbon containing aerosol. If available, a direct comparison between the SP-AMS and the MAAP during calibration with representative BC materials may be useful here.**
    We do agree with the Referee regarding this approach, however the suggested comparison is not available but we will take this into consideration for future work.

13. **L245: How much of an effect does scaling SP-AMS rBC to the MAAP have on the comparison between AMS total PM1 and SMPS PM1?**
    This is an interesting point, which we have investigated further. Scaling of rBC to $BC_{MAAP}$ data have an effect of around 10% and can therefore not explain the difference between the two instruments.

14. **L247-250: During the beginning few days of the study the SMPS concentration is higher than the SP-AMS concentration, and this is the only period where this appears to be the case. SP-AMS chloride was elevated during this time; was any measurable signal for NaCl+ present at that time?**
    The authors find the identification of NaCl interesting but at the same time beyond the scope of this manuscript, which is why this is not investigated any further.

15. **L291-293: It may be worth clarifying this. Long range transport suggests to me sources far outside the Arctic, but the authors suggest that this is likely not the case after the transition to cleaner conditions. Or, do the authors suspect that the MOA was transported from more southerly marine regions? It may be useful to provide some general indication of the meteorological regime or air mass histories, for example, for before and after the Arctic Haze decreases substantially at Villum.** We agree that the transport issue appears somewhat unclear. The dominating wind direction is southwestern with an average wind speed of 4 m s$^{-1}$ as explained in section 2.1. In the introduction section we show that Eurasia is the dominating source region during winter, i.e. line 73-75: *"Due to the expansion of the polar dome, a major part of the aerosol mass is long-range transported from source regions outside the Arctic where the primary source region has been identified as the northern part of Eurasia (Nguyen et al., 2013; Quinn et al., 2008; Heidam et al., 2004; Stohl et al., 2007; Christensen, 1997; Abbatt et al., 2019)."*

    Furthermore, we have rewritten the following sentence: Lines 303-309: *"As expected no diurnal pattern is observed for any of the chemical species. These are mainly transported from long distances. For example, the source regions that contributed to ground-level $SO_X$ at VRS were assigned to Western Europe (7%), Eastern Europe (9%), Asia (2%), North America (7%) and Russia being the main emitter by far (75%) (Heidam et al., 2004). During summer, the atmospheric circulation is confined within the Arctic region and is considered essentially local. Thus, marine biogenic sources that peak during spring and summer are expected to origin from within the region."*

16. **L300-301: Long range transport from marine regions can mean that a portion of springtime Arctic sulphate is from DMS oxidation (https://www.atmos-chem-phys.net/17/8757/2017/)**
    We thank the Referee for this comment and we have rewritten the sentence and included the reference: Line 322-323: *"Secondary $SO_4^{2-}$ is formed by atmospheric oxidation of sulfur dioxide ($SO_2$) and to some extent DMS ..."* and line 333-335: *"However, a recent study using both airplane measurements and modeling suggest that long-range transport of DMS is significant during spring (Ghahremaninezhad et al., 2017)."*

17. **L313: Is the contribution of MSA subtracted from reported organics and sulphate?**
    We have not subtracted MSA from the reported organics or sulphate.

18. **L315: Is this a statement based only on the measurement period presented in this study. Measurements of MSA at Alert would suggest that DMS peaks later in the summer (Leaitch et al., Elementa, 2013: https://www.elementascience.org/articles/10.12952/journal.elementa.000017/)**
    Unfortunately, a typo has sneaked in. We have replaced DMS with MSA and the sentence in line 339-340 now reads: *"In this study, MSA emerges steadily and peaks the end of April (see Section 3.2)."*

19. **L331: Biomass burning may be a larger source to the Arctic than farms**
    We thank the Referee for the comment and have correct the sentence: Line 353-356: *"In contrast, ammonia ($NH_3$) which is the precursor of $NH_4^+$, derives largely in winter and spring from long-range transport of emissions from biomass burning and agriculture (Fisher et al., 2011), whereas in summertime $NH_3$ emission from seabird-colonies can play a significant role (Croft et al., 2016)."*

20. **L342: Current evidence suggests that frost flowers may not be an important source of aerosol (see for example, Huang 2017 https://www.atmos-chem- phys.net/17/3699/2017/ and references within)**
    We thank the Referee for the comment and added this information to the text: Line 372-373: *"Current research has suggested that blowing snow might be a much more dominant source of sea salt aerosols compared to frost flowers (Huang and Jaegle, 2017)."*

21. **L351-353: This type of information may belong more in the introduction, rather than the results and discussion**

This is correct and we have therefore deleted the sentence from this paragraph and added a small part of it in the following sentence: Line 380-383: *"These accumulation mode BC-containing particles can be transported over longer distances during the Arctic haze period and may serve as cloud seeds in the late spring, when precipitation begins to be important in the Arctic (Bond et al., 2013; AMAP, 2011; Massling et al., 2015; Garrett et al., 2011)."*

22. **L364-366: The authors need to screen the data based on wind direction, or another measured parameter, before reporting and interpreting R2 values here.**
These considerations are discussed in section 3.2, where we conclude that local contamination accounts for less than 1%.

23. **L385-395: A mean pToF size distribution for the Arctic Haze period and the more biogenically influenced period could help with this interpretation of mixing state (and would be very interesting!).**
We agree that this could be interesting but it is not possible to investigate any further with the current data set.

24. **L406: Secondary or highly aged primary organic aerosol, it is difficult to interpret a mechanism based solely on CO2+ contribution alone. I suggest the authors elaborate on how they reached this conclusion**
We thank the Referee for the comment and we acknowledge that this needs further clarification. Lines 434: *"The $CO_2^+$ ion ...likely secondary in origin"* has been deleted. A new paragraph was inserted instead: Line 431-439: *"Oxygenated aerosols from numerous field campaigns on the northern hemisphere are deconvolved into HOA and OOA. OOA has been shown to account for a large fraction of OA and to be a good surrogate for secondary organic aerosols (SOA) in multiple studies (Ng et al., 2010, Zhang et al., 2007, Zhang et al., 2011). Oxygen containing functional groups produce m/z 43 ($C_2H_3O^+$) and m/z 44 ($CO_2^+$) fragments, which are prominent peaks in OOA mass spectra (Ng et al., 2010), including those of MOA and AOA found in this study. These factors are highly OOA factors with O/C ratios of 0.63 and 0.95, respectively. According to Jimenez et al. (2009) these factors would be classified as low volatility OOA (LV-OOA). There is strong evidence that OOA is secondary in nature and several studies of aging indicate that OA converges towards LV-OOA following numerous steps of atmospheric oxidation (Jimenez et al., 2009)."*

Furthermore, the following sentence has been changed to: Line 424-434: *"The dominating OA appears to origin from long-range transport into the region during winter/spring."*

25. **L412: It might be useful to indicate the significance of these correlations in Table 2 in some way (e.g., bold R2 values)**
We thank the Referee for the suggestion, however the great amount of data can be problematic in a t-test and can lead to misleading conclusions if it is not described any further in the manuscript. We therefore prefer to describe the significance of the correlations (where it is relevant) in the text and not in Table 2.

26. **L441-442: It is really difficult to attribute primary or secondary sources from the mass spectrum alone. Marine OA observed at Mace Head is likely a combination of primary and secondary OA. Ovadnevaite et al., GRL 2011 (https://doi.org/10.1029/2010GL046083) state (paragraph 9): "The relatively high amount of oxygenated organics typically indicates a chemical aging of the aerosol [Jimenez et al., 2009] with possible contributions from both oxidation of primary aerosol organics and SOA (secondary organic aerosol) formation."**
We completely agree with the Referee and have tried to make the sentence clearer so that is describes exactly this: Line 477-480: *"The resemblance of MOA from this study with the mass spectrum from Mace Head and the high O/C ratio of 0.95 indicate, that MOA is composed of chemically aged aerosols from both oxidation of primary aerosols and secondary organic aerosols (Ovadnevaite et al., 2011; Fu et al., 2015)."*

27. **L 454-455: "In April, the highest OA concentrations is observed where AOA accounts for around 70% of OA (Figure 4). In May, MOA becomes the dominating OA while AOA nearly disappears." This seems to be an important point of the paper, which could be further highlighted in the abstract/conclusions and results & discussion sections.**
We have tried to make it more evident in the abstract: Line 37-39: *"The MOA emerged late at the end of March, where it increased with solar radiation and reduced sea ice extent, and dominated OA for the rest of the campaign until the end of May (24-74% of OA), **while AOA was nearly absent**."*

28. **L456-457: While comparing to this Alert study is valuable, Narukawa use a very different method and their data represent measurements from 15 years prior to this study. A direct comparison is difficult to make, but I agree it is interesting despite these differences.**
We thank the Referee for this valuable comment and we have added more information to the sentences: Line 494-497: *"This is significantly higher than observed at Alert by Narukawa et al. (2008) where marine organic matter contributed 45% to aerosol total carbon in late spring (26 April – 6 May 200). However, direct comparison is difficult due to different methods and time periods (Narukawa et al., 2008)."*

29. **L461-462: Some additional references related to DMS would be useful here. The marginal ice zone is also important for DMS**
A paragraph has been in added in line 502-506: *"This can be visualized from the strong coupling between DMS concentration and chlorophyll-a from DMS producing phytoplankton (Park et al, 2013). Moreover, Becagli et al (2016) concluded that oceanic primary production was related to melting of sea ice and extension of marginal sea ice areas based on satellite derived chlorophyll-a and measurements of MSA (Becagli et al., 2016)."*

30. **L465-471: Some more detailed information about the source regions impacting Villum during winter and spring might help this discussion and interpretation.**
We hope that the information provided in comment no. 15 serves as background information for the sentence, which this comment refers to.

31. **L473-474: Why speculate about the emergence of a continental factor?**
We agree that this is unnecessary and have deleted this part of the sentence (line 517).

32. **L474: Reporting an overall average MOA fraction here is a bit confusing, since the previous discussion demonstrates its much higher contribution once the AOA decreases.**
This is a very good point and we have therefore added the changing mass fraction of MOA over the measurement period: Line 518-519: *"...ranging from 2-3% of OA in February and March to 24% and 74% of OA in April and May, respectively (Figure 2b and 4)."*

33. **L475-477: In addition, and perhaps more importantly MOA dominates the organic aerosol mass when the overall concentrations are very low, particle numbers are low, and so cloud condensation nuclei concentrations can be low.**
We thank the Referee for this valuable comment and have therefore changed the sentence to accommodate the suggestions: Line 517-522: *"MOA constitutes 22% of OA on average during our measurement period ranging from 2-3% of OA in February and March to 24% and 74% of OA in April and May, respectively (Figure 2b and 4). Thus, MOA is by far the most abundant OA from end of April and onwards. MOA dominates the OA mass after polar sunrise and persists during polar daytime so the aerosol's optical impact might be substantial. At the same time, MOA dominates when the overall $PM_1$ concentration is very low, particle numbers are low and hence CCN concentrations can be low."*

34. **L480-481: Oxidation of DMS and other VOCs would be considered secondary. The wording of this sentence is a bit confusing**

We thank the Referee for finding this mistake and we have now corrected the sentence: Line 524-527: *"MOA may contain oxidation products of DMS and other VOCs from oceanic origin, as well as a variety of primary components including sacharides such as mannitol in addition to insoluble gels (Croft et al., 2018; Fu et al., 2013, Leck and Bigg, 2005; Orellana et al., 2011; Ovadnevaite et al., 2011)."*

35. **L482-483: In Croft ACP 2019, secondary OA accounted for up to half of the summer-time OA, and primary marine OA also contributed. The authors may want to be more clear in their usage of marine OA, primary OA and secondary OA. Marine OA can come from both primary and secondary processes.**
A paragraph has been inserted: Line 480-481: *"Aerosol growth has been correlated with the presence of MSA, and other organic species (Willis et al., 2016)."*

Also, the last paragraph in the result section have been modified: Line 524-528: *"MOA may contain oxidation products of DMS and other VOCs from oceanic origin, as well as a variety of primary components including sacharides such as mannitol in addition to insoluble gels (Croft et al., 2018; Fu et al., 2013, Leck and Bigg, 2005; Orellana et al., 2011; Ovadnevaite et al., 2011). In line with our findings, modelling at several sites in the Canadian Arctic suggested that marine OA other than MSA may account for more than half of the summertime OA (Croft et al., 2018)."*

36. **L484-487: This introductory information may fit better within the introduction.**
We have modified and moved this part to the introduction where it fits better: Line 114-117: *"Biogenic marine aerosols can scatter solar radiation, which will result in a negative radiative forcing. Biogenic marine aerosols can also coat soot particles, which may be transported from wild fires (AMAP, 2015), which could impact the CCN activity and absorption by the soot particles (Lange et al., 2018)."*

37. **L493-495: Comparing to Alert may also be warranted, given its proximity to VRS**
A sentence has been added to section 3.1: Lines 313-316: *"Sulfate is dominated by anthropogenic sources accounting for 65% at Alert (Norman et al., 1999) and 75% Svalbard (Udisti et al., 2016) as annual averages. On the contrary, biogenic sources accounted for 63% of sulfate in size fraction smaller than 490 nm at Alert during summer (Ghahremaninezhad et al., 2016)."*

Furthermore, in the conclusion we have added the following: Line 535-537: *"This is in accordance with previous findings from VRS, Alert (Norman et al., 1999) and Svalbard (Udisti et al., 2016) where $SO_4^{2-}$ has been apportioned to be 65% and 75% anthropogenic, respectively."*

38. **L495-500: The authors' clear observations of changing OA character and sources over the winter to spring to late spring transition may be a more important conclusion that that these organic species can be mixed with rBC.**
We agree with Referee and have removed the sentence from the conclusion (line 541).

39. **L506-507: The observations presented here cannot unambiguously determine whether AOA and MOA is primary or secondary in origin. The mass spectrum similar to Ovadnevaite 2011, only suggests that the aerosol is marine in origin. More information would be needed to suggest a dominant formation process. While the correlation of AOA with sulphate may suggest secondary processes, this aerosol is also transported over very long distances and so aerosol from somewhat different formation processes may co-vary in time at such a remote location.**
This is true. Please also refer to comment no. 24 and 26. We have rephrased the sentence: Line 547-548: *"AOA and MOA showed evidence of SOA. Furthermore, the resemblance of MOA with a previously published marine organic plume where indicative of MOA having a primary organic component."*

40. **L512-514: I agree in general with this statement, but some more information about source regions impacting Villum would go a long way in this interpretation. Further, do the authors have access to CO data that could potentially help to demonstrate the increase in deposition mentioned here? (e.g., see Garrett et al., GRL, 2011 doi:10.1029/2011GL048221)**

We acknowledge this comment, which is very much in line with comments no. 15 and 30. In line 555-556, we inserted a sentence: *"This is supported by the confined atmospheric circulation within the Arctic region during summer (Heidam et al., 2004)."* Unfortunately no CO data is available for campaign period.

41. **Figure 3: That the authors observe a distinct HOA factor in Arctic haze that co-varies in time more closely with rBC than with AOA or sulphate is interesting. Intuitively I would expect Arctic haze aerosol to be overall extremely oxidized, though the prevalence of HOA in the dark winter suggests not. Do the authors have specific evidence to show that the HOA was not more regionally sourced than the AOA? Do polar plots of wind direction/speed and PNF factor intensity shed any light on differences in source regions?**
We thank the Referee for the comment but we do not have specific evidence to show whether the HOA factor is more regionally sourced than the AOA factor. However, the less oxidized state of HOA does suggest that HOA could be more regionally sourced than for example AOA.

42. **Figure 4: Does the MSA-to-sulphate ratio, and organic-to-sulphate ratio, increase in a similar manner to MOA on a monthly basis?**
Yes, the MSA-to-$SO_4$ ratio does increase in a very similar pattern to MOA on a monthly basis. The OA-to-$SO_4$ ratio is relatively constant from February to May, which is also evident from Figure S6 (line S120).

[revised manuscript text omitted]

---

## Author Comment (AC2) · 12 Jul 2019

**Referee #3**

1. **Lines 24 – 25: Do "organic matter" and "organic aerosol" both refer to organic aerosol concentrations as ug C/m3 or as total particulate organic matter including H and O?**
   We refer to organic aerosol concentration as total particulate organic matter and we have therefore changed the sentence to: Line 26-27: *"The second most abundant species was organic aerosol (OA) (24%)."*

2. **Lines 78 – 79: Decreasing trends in nss SO4 and BC have been documented for Barrow. Please see Chapter 9 of the 2015 AMAP report on Black Carbon and ozone as Arctic climate forcers ([www.amap.no](www.amap.no)).**
   We thank the Referee for this important comment. We have deleted the original sentence and changed the following sentence to include Barrow and a new reference: Line 83-85: *"Since then, $SO_4^{2-}$ and BC during winter-spring have declined at Alert, Mount Zeppelin, Barrow and VRS (Heidam et al., 1999; Hirdman et al., 2010; AMAP, 2015)."*

3. **Line 214: Applying a uniform specific absorption coefficient for BC could affect temporal variability if the nature of the BC (source, aging processes, etc.) lead to varying specific absorption coefficients.**
   We thank the Referee for this important point and have corrected the sentence: Line 222-223: *"Uncertainty in the conversion factor likely impacts the reported absolute concentrations, and potentially the temporal variability."*

4. **Lines 248 – 249 and SI lines 85 – 98: It is not clear from the main text that periods where differences between PM1 determined from the SP-AMS and the SMPS were at least 2 ug/m3 (late March/early April and mid-April) were excluded from the data analysis. It states in the SI that data from Feb 21 – 26 and Mar 29 – Apr 2 were excluded. Please clarify this in the main text. Also – what is the impact of not including sea salt in the SP-AMS derived PM1 since it will be included in the SMPS PM1? The modal number diameter of the sea salt mode is ←-200 to 300 nm so should be detected by the SMPS.**
   No data has been excluded based on the data comparison with SMPS. The text in SI states that the data was not excluded despite the poor correlation with the SMPS: Line S101-103: *"No explanation could be found for the relatively poor correlation in the beginning of the campaign (21-26 February) and in the end of March (29 March – 2 April), which is why data has not been excluded."*

   The question concerning sea salt and the instrument comparison is interesting but currently we have no data available regarding sea salt during the campaign and therefore it is unfortunately not within the scope of the work.

5. **Lines 312 - 315: What is the MSA to SO4 ratio during periods when MSA was detected? Can the ratio be used to assess the importance of biogenic vs. anthropogenic sources of SO4?**
   This is an interesting question. In Nguyen et al. (2013), we apportioned 7-9% of $SO_x$ to the marine source as an annual average using PMF and COPREM and inorganic species ($PM_{10}$). In this study, we will most likely always have an unknown anthropogenic sulfate contribution in the summer like the 35% anthropogenic sulfate identified by Ghahremaninezhad et al. (2017) during summer at Alert. That is, the pure $MSA/SO_4$ (marine) will likely not be possible to identify.

6. **Line 340 – 342: Is the attribution of Cl and NO3 to frost flowers (i.e., a local source) due to their presence in the supermicron size range? Please clarify in the main text.**
   We thank the Referee for the comment and have clarified the text: Line 368-370: *"
[revised manuscript text omitted]

---

## Author Comment (AC3) · 12 Jul 2019

**Referee #2**

**The study reports on SP-HR-AMS measurements conducted at Villum Research Station in the north of Greenland from February to May 2015. The authors investigate the concentrations and evolution of refractory black carbon (rBC), particulate sulfate (SO4) and organic aerosol (OA). The first half of the manuscript focuses on rBC, the second on OA that was further investigated by conducting positive matrix factorization (PMF). Three factors were identified: hydrocarbon-like OA (HOA) with the smallest contribution, Arctic haze OA (AOA) with the largest contribution and marine OA (MOA).**

**Detailed measurements of rBC and OA in the high Arctic are rare, especially outside of the summer season. The real strength of this study are the real-time observations during the transition period from winter to spring when sunlight returns and Arctic haze conditions fade. While the authors make this point, they also "dilute" their message by putting emphasis on reporting average concentrations for the entire study period, which do not address the environmental change. Generally, this study provides valuable insights into the aerosol chemical composition in the high Arctic and should be published with major revisions as suggested below. General and specific comments are mentioned below, all other comments are highlighted in the attachment.**

**General comments:**
**A shortcoming of the study is that it underexplores the HR-AMS data. There is no reporting of hetero-atoms such as nitrogen or sulfur in the OA. The contribution of those as a function of time could reveal more details about the sources of MOA in particular. At the moment only O:C ratios are provided. I suggest exploring also the N- and S-containing contributions to OA. In particular the contribution of MSA should be quantified. MSA is discussed in the manuscript (l. 437ff), but rather superficially. See also respective comment in the manuscript.**

We thank the Referee for the comment and we have indeed scanned the data for amines and did not find anything interesting.

**The authors mention often the average concentrations of the constituents during the campaign. As mentioned above the real strength of the observations lies in having captured the transition periods and the transition cannot be described by campaign average but should rather be discussed as gradients are differences. How long does the transition take, which markers change first, which ones later, or all simultaneously? I suggest changing the emphasis to transition characterization throughout the whole manuscript. For example: l. 345: here an average BC concentration is mentioned; l. 367: a slope or gradient for the SO4 concentration would make more sense here;**

This is a very sound and valid comment and we thank the Referee for the suggestion. Referee no. 1 had similar comments and we have therefore now removed the average concentrations from the abstract and rewritten some of the manuscript to accommodate these comments: Line 23-39:

"*During this period, we observed the Arctic haze phenomenon with elevated PM$_1$ concentration ranging from an average of 2.3, 2.3 and 3.3 µg m$^{-3}$ in February, March and April to 1.2 µg m$^{-3}$ in May. Particulate sulfate (SO$_4^{2-}$) accounted for 66% of the non-refractory PM$_1$ with highest concentration until the end of April and decreasing in May. The second most abundant species was organic aerosol (OA) (24%). Both OA and PM$_1$, estimated from the sum of all collected species, showed a marked decrease throughout May in accordance with the polar front moving North together with changes in aerosol removal processes. The highest refractory black carbon (rBC) concentrations were found in the first month of the campaign averaging 0.2 µg/m$^3$. In March and April, rBC averaged 0.1 µg/m$^3$ while decreasing to 0.02 µg/m$^3$ in May.*

*Positive Matrix Factorization (PMF) of the OA mass spectra yielded three factors: (1) a Hydrocarbon-like Organic Aerosol (HOA) factor, which was dominated by primary aerosols and accounted for 12% of OA mass; (2) an Arctic haze Organic Aerosol (AOA) factor; and (3) a more oxygenated Marine Organic Aerosol (MOA) factor. AOA dominated until mid-April (64%-81% of OA), while being nearly absent from the end of May and correlated significantly with SO$_4^{2-}$, suggesting the main part of that factor being secondary OA. The MOA emerged late at the end of March, where it increased with solar radiation and reduced sea ice extent, and*

*dominated OA for the rest of the campaign until the end of May **(24-74% of OA), while AOA was nearly absent.***"

In addition, we have gone through the entire manuscript and corrected/changed paragraphs where average campaign concentrations were presented. Changes are shown below:

- Line 294-296: We added ranges to *"The total measured $PM_1$ concentration during the field study may seem relatively high, averaging 2.3 μg $m^{-3}$**- ranging from 2.3, 2.3 and 3.3 μg $m^{-3}$ in February, March and April to 1.2 μg $m^{-3}$ in May**."*
- Line 317-319: We deleted the average $SO_4^{2-}$ concentration and changed it to: *"During the entire campaign, $SO_4^{2-}$ is the dominant species that on average makes up almost 70% of the $PM_1$ mass concentration **with highest concentration until the end of April and decreasing in May (Figure 1b-c).**"*
- Line 342-343: We deleted the average OA concentration and changed it to: *"In this study, the OA fraction is the second largest contributor to $PM_1$ **where** weekly averages showed a clear decrease from mid-April relative **to concentrations in February and March** concentrations (Figure 1)."*
- Line 349-350: We deleted the average concentration and changed it to: *"Particulate $NH_4^+$ is found in much lower concentrations compared to OA and $SO_4^{2-}$ **but with the same transition pattern as the two other species.**"*
- Line 361-362: We have kept the mentioning of the average concentration of $NO_3^-$ and $Cl^-$ due the comparison with the detection limit of the species.
- Line 374-375: We have deleted the campaign average for rBC and instead looked at concentration for the different months: *"The highest rBC loadings are found in the first month of the campaign (February) **averaging 0.2 μg/$m^3$. In March and April, the average is 0.1 μg/$m^3$ which then decreases to 0.02 μg/$m^3$ in May.**"*
- Line 440-442: We have added more information concerning the AOA development: *"AOA accounts for 64% of OA mass for the entire field study **but ranges from 64%, 81% and 71% of OA in February, March and April to 20% in May (Figure 2b and 4).**"*
- Line 517-519: We have added more information concerning the MOA development: *"MOA constitutes 22% of OA on average during our measurement period **ranging from 2-3% of OA in February and March to 24% and 74% of OA in April and May, respectively** (Figure 2b and 4)."*
- Line 550-554: We have added more information concerning the AOA and MOA development in the conclusion: *"The less oxidized AOA builds up during the Arctic haze period and dominates until early spring **(64%-81% of OA),** during which both the absolute and relative contribution to the OA burden decreases substantially. In contrast, the MOA is nearly non-existent until early spring but is then by far the dominating OA from the end of April and onwards **(24-74% of OA).**"*
- In regard to line 367 (now line 394-395) we acknowledge the Referee's opinion but we believe the two ways of presenting this are equivalent. We have therefore kept the original sentence and not used a slope or gradient as suggested by the Referee.

Finally, we have changed the first paragraph of the conclusion to accommodate this comment (marked in bold): Line 532-541:

*"In the transition from polar night to polar **day we observed elevated $PM_1$ concentration ranging from an average of 2.3, 2.3 and 3.3 μg m-3 in February, March and April to 1.2 μg m-3 in May.** We concluded SO42- to be the most abundant species in sub-micrometer aerosols **with highest concentration until the end of April and decreasing in May.** This is in accordance with previous findings from VRS, Alert (Norman et al., 1999) and Svalbard (Udisti et al., 2016) where SO42- has been apportioned to be 65% and 75% anthropogenic, respectively. **While not previously quantified at VRS, OA was found to be the second largest contributor to PM1 (24%). As for the other species, OA showed a decrease in concentration from mid-April relative to February and March. rBC concentration were found to be highest in the first month and then decreased throughout the campaign – average concentration of 0.2, 0.1, 0.1 and 0.02 μg m-3 in February, March, April and May, respectively.**"*

**I suggest renaming the title to "Biogenic and Anthropogenic sources of Arctic Aerosols at Villum Research Station". "Arctic Aerosols" alone is misleading, because the measurements reflect the unique environment of VRS in northern Greenland. That is very different from the Canadian archipelago or Svalbard as the authors write themselves.**

We thank the Referee for this valuable comment, which was also suggested by Referee no. 1. To accommodate both Referees we have changed the title to: *"Biogenic and anthropogenic sources of aerosols at the high Arctic site Villum Research Station".*

**Along the same line is the inaccuracy with which the authors cite literature in the introduction:**

1. **L. 37: How do the authors define the "Arctic summer aerosols"? Do they mean the high Arctic, so basically the Arctic Ocean? Or do they include terrestrial parts of the Arctic. This makes a fundamental difference for the composition and other properties of aerosols.**

   We thank the Referee for the comment and have changed the sentence to: Line 40-42: *"Our data supports current understanding that Arctic aerosols are highly influenced by secondary aerosol formation, and with an important contribution from marine emissions during Arctic spring in remote high Arctic areas."*

2. **L. 79: This information is incomplete. The paper also states that SO4 decreased significantly in Alert and Zeppelin and that the lack of a trend at Barrow is likely due to the limited data coverage. This information needs to be added.**

   We thank the Referee for this important comment. We have deleted the original sentence and changed the following sentence to include Barrow and a new reference: Line 83-84: *"Since then, $SO_4^{2-}$ and BC during winter-spring have declined at Alert, Mount Zeppelin, Barrow and VRS (Heidam et al., 1999; Hirdman et al., 2010; AMAP, 2015)."*

3. **L. 86: This article is focused on the Canadian Arctic mostly. Use literature that is more relevant to the entire Arctic. Furthermore, the article has been published in 2019 in ACP.**

   We have updated the sentence and references therein: Line 92-93: *"Transport reaches a minimum in late spring where wet deposition becomes an important removal process (Abbatt et al., 2019; AMAP, 2015)."*

4. **L. 112 "DMS emissions in the Arctic have increased by 30 %..." Is this true for the entire Arctic or the Canadian sector? It is important to provide a differentiated picture of what is happening, otherwise false impressions are created.**

   Based on Abbatt et al., 2019, the increase is valid for Arctic and based on a new satellite-based model. To clarify this, we have revised the sentence: Line 121-122: *"A new satellite-based model suggests that DMS emissions in the Arctic have increased by 30% per decade the last two decades due to both increased temperatures and decreased ice cover (Abbatt et al., 2019)."*

5. **L. 114: "demonstrated" is an overstatement, the paper infers. The authors show the relationship but do not provide an explanation.**

   We have changed the sentence accordingly: Line 123-124: *"A relationship between MSA and the frequency of new particle formation has also been inferred based on long-term observations (Dall'Osto et al., 2017)."*

6. **L. 115: MSA does not nucleate or form new particles, it rather condenses and grows particles.**

   To make this clearer we have revised the sentence: Line 123-125: *"A relationship between MSA and the frequency of new particle formation has also been inferred based on long-term observations (Dall'Osto et al., 2017) although MSA cannot be the nucleating part."*

7. **L. 116: It is not only believed that ammonia comes from sea bird colonies, this has been shown multiple times. There are global inventories for ammonia seabird emissions even.**

This is correct and we have changed the sentence to: Line 125-127: "*Another important natural source of Arctic aerosols is ammonia, which among other things is believed to originate from migrating sea bird colonies (Croft et al., 2016).*"

**Specific comments:**

8. **L. 23: unclear whether the particulate sulfate or PM1 amounted to 2.3 ug / m3**
   In order to accommodate the Referee's comment concerning the use of average concentrations we have changed this sentence completely. At the same time, we have made it clearer in regard to sulfate and $PM_1$: Line 23-26: "*During this period, we observed the Arctic haze phenomenon with elevated $PM_1$ concentration ranging from an average of 2.3, 2.3 and 3.3 µg $m^{-3}$ in February, March and April to 1.2 µg $m^{-3}$ in May. Particulate sulfate ($SO_4^{2-}$) accounted for 66% of the non-refractory $PM_1$ with highest concentration until the end of April and decreasing in May.*"

9. **l. 40: Why is it urgently needed to elucidate the chemical components? The authors probably mean that modeling the future of the Arctic requires process understanding. Just because climate is changing doesn't mean we need highly time resolved aerosol data.**
   We thank the Referee for the comment and we have changed the sentence to: Line 42-44: "*In view of a changing Arctic climate with changing sea-ice extent, biogenic processes, and corresponding source strengths, highly time-resolved data are needed in order to elucidate the components dominating aerosol concentrations to enhance the understanding of the processes taking place.*"

10. **l. 45: consider referring to the special IPCC report on 1.5 C and the AMAP 2015 report on BC and ozone in the Arctic.**
    We thank the Referee for the suggestion and have now changed the references to IPCC, 2018 and AMAP, 2015 and revised the sentences: Line 47-49: "*Climate change driven by anthropogenic emission of greenhouse gases seriously impacts the Arctic, which has experienced average temperature increases of twice the global mean during the last 100 years (AMAP, 2015; IPCC, 2018).*"

11. **l. 52: ice does not condense onto particles**
    We have changed the sentence accordingly: Line 56: "*…by serving as cloud-condensation and ice nuclei*".

12. **l. 63: Is it truly "visible"? Strong haze events might be visible by eye, but the typical Arctic Haze is still orders of magnitude lower in mass concentrations as the visible urban air pollution, as is somehow inferred by this sentence.**
    We have changed the sentence to: Line 70-71: "*The Arctic haze peaks in early spring (Heidam et al., 1999; Law and Stohl, 2007; Stohl, 2006; Heidam et al., 2004; Abbatt et al., 2019).*"

13. **l. 67: As it is written it contradicts above statement that says that Arctic Haze sources are located within in the polar dome. This needs some clarification or more exact formulation.**
    To make the formulation clear we have changed the sentence to: Line 73-75: "*Due to the expansion of the polar dome, a major part of the aerosol mass is long-range transported from source regions outside the Arctic where the primary source region has been identified as the northern part of Eurasia.*"

14. **l. 87: why should vegetation fires not be considerable? It's a question of whether their emissions are transported to the high Arctic.**
    To make it more precise we have removed "still" from the sentence: Line 73-74: "*Natural emissions from vegetation fires can be considerable in spring and early summer (Mahmood et al., 2016).*"

15. **l. 93: Consider referring also to Chang et al., 2011, ACP doi:10.5194/acp-11-10619- 2011 They characterize PM1 aerosol measured with an AMS and PMF in the central Arctic during the ASCOS campaign. Also Willis et al., 2018, 10.1029/2018RG000602 provide an overview of what we know about Arctic aerosol and it's detailed composition.**

Thank you for the suggestion and we have now added the reference: Line 100-102: *"…though few studies have characterized this component in detail (Barrett et al., 2015; Brock et al., 2011; Frossard et al., 2011; Kawamura et al., 2010; Quinn et al., 2002; Shaw et al., 2010; Leaitch et al., 2018; Chang et al., 2011; Willis et al., 2018)."*

16. **l. 98 ff: This seems to be more a concluding statement which should be placed later. It is a bit awkward after the OA discussion.**
We thank the Referee for the input, and we have moved the sentence to another paragraph (line 56-59) where it fits better.

17. **l. 108: the explanation why the role is important is missing.**
We have revised the sentence: Line 113-114: *"Marine aerosols play an important role for the climate due to their optical properties and ability to alter cloud nucleation (Abbatt et al., 2019; Willis et al., 2018)."*

18. **l. 110: Unclear where MSA is increasing.**
Correct and after re-reading the references we have revised the sentence: Line 119-120: *"MSA levels have been associated with marginal sea ice moving North."*

19. **l. 123: revise the sentence, it is grammatically incorrect and does not list the two disadvantages.**
We have revised the sentence: Line 133-134: *"Beside the low time resolution, a disadvantage of these types of measurements can be evaporate loss or adsorption of semi-volatile compounds."*

20. **l. 126: delete "and trends". Trends are longer term changes.**
We have deleted "and trends" (Line 136).

21. **l. 139: PMF cannot reveal source regions just source types.**
We have now corrected the sentence: Line 148: *"…and to allocate potential sources and source types."*

22. **l. 153: Where is the HVS data used? This is not evident in the manuscript. If they are used that needs to be stated and then more information like flowrate, sample duration etc. needs to be added, or a reference to the supplement needs to be given.**
The HVS data is used to collect filter samples of EC and OC and the information on flow rate and sample duration is already presented in the supplement. To make this clear we have added the following sentence in the manuscript: Line 162-163: *"More information concerning the supplementary instruments can be found in Supporting Information."*

23. **l. 176: "inspected" sounds like the flow rate was measured once. I hope it was checked several times during the campaign.**
We have rephrased the sentence: Line 184-185: *"The flow rate was controlled regularly with a Gilian Gilibrator… ."*

24. **l. 176 if the size calibration was conducted with ammonium nitrate, a DMA must have been operated as well to select a range of sizes. This information is missing entirely.**
We have deleted this part of the sentence since this is not relevant for this manuscript (line 185).

25. **l. 179: Why was there no determination of the relative ionization efficiency of sulfate with ammonium sulfate?**
In an ideal campaign we should have carried out ammonium sulfate calibration as well. Thus, the Referee's comment has been noted for future campaigns.

26. **l. 191: The AMS also sees NaCl, see Ovadnevaite et al., 2012, doi:10.1029/2011JD017379. and other publications. The influence of NaCl needs to be considered as well.**

This is an interesting point but at the same time it is beyond the scope of this manuscript which it focusing on the organics. We have added the reference to Line 363-366: *"However, the SP-AMS does not typically measure refractory chloride at normal vaporizer temperatures, such as NaCl (Canagaratna et al., 2007). Although, Ovadnevaite et al. (2012) has demonstrated how the AMS could be calibrated to measure NaCl in high-time resolution."*

**27. l. 214: add manufacturer and model number of the SMPS.**
We have addressed this comment by adding the following sentence: Line 225-226: *"The SMPS is custom-built with a Vienna-type medium column and more information can be found in Lange et al. (2018)."*

**28. l. 224: "majority". Can the authors be more specific and provide the quantiles?**
We have added percentage to the sentence: Line 234: *"The time dependent CE varied with the majority (> 97%) of values between 0.8 and 1…"*

**29. l. 253: the sentence is confusing.**
We agree and have re-written the sentence: Line 263-265: *"PMF analysis (Paatero, 1997; Paatero and Tapper, 1994; Lanz et al., 2007; Ulbrich et al., 2009) was conducted on the time dependent organic mass spectra to determine OA factors and potential sources of OA."*

**30. l. 273: "chemical composition" instead of "chemistry"**
As requested, we have replaced *"chemistry"* with *"chemical composition"* (line 283).

**31. l. 285: A comparison to other studies is missing that would reveal why the concentration can be perceived as relatively high.**
The sentence has been modified to: Line 294-296: *"The total measured $PM_1$ concentration during the field study may seem relatively high, averaging 2.3 μg $m^{-3}$ - ranging from 2.3, 2.3 and 3.3 μg $m^{-3}$ in February, March and April to 1.2 μg $m^{-3}$ in May."*

Furthermore, a paragraph has been added: Line 309-316: *"Arctic sites show similar increases in key particulate pollutants in winter and early spring, where maximum sulfate concentrations may reach 3 μg $m^{-3}$ as compared to average summer concentrations of 0.1 μg $m^{-3}$ (Quinn et al., 2007). For example, typical PM1 concentrations were 0.1 - 0.2 μg $m^{-3}$ in August to September during the ASCOS expedition (Chang et al., 2011). Sulfate is dominated by anthropogenic sources accounting for 65% at Alert (Norman et al., 1999) and 75% Svalbard (Udisti et al., 2016) as annual averages. On the contrary, biogenic sources accounted for 63% of sulfate in size fraction smaller than 490 nm at Alert during summer (Ghahremaninezhad et al., 2016)."*

**32. l. 302: What is the role of light here?**
Although OH can be formed in dark reactions, photolysis of $O_3$ and subsequently reaction with $H_2O$ is the dominating source of OH.

**33. l. 303: "at its source region" This should rather read: "in the vicinity of the source region, "SO2 oxidation does not happen immediately and normally SO2 has already been transported away some distance from the source before it is oxidized to SO4 2-**
We agree and have changed the sentence to: Line 324-325: *"Secondary long-range transported $SO_4^{2-}$ depends on atmospheric oxidation of $SO_2$ at the vicinity of the source regions…."*

**34. l. 305: Figure 3 is mentioned before Figure 2.**
We have corrected this and there is now a mention of Figure 2 before Figure 3 in line 278.

**35. l. 308: "originating from Siberia" Is this not a contradiction to the main wind direction from the south-west? How representative is the wind direction of the general atmospheric circulation around VRS?**

This is a very important point because the wind rose presented in Figure S1 is only representative for the wind direction at ground level at the measurement site. This information is primarily used in the manuscript in regard to identifying local pollution from the military station located 3 km from the measurement site. Hence, it cannot be used for interpreting anything general regarding transport direction or emission areas. For this, air mass back-trajectories should be applied as is the case in Nguyen et al., 2013, which shows change in wind directions and source areas at different altitudes.

36. **l. 319: How do you define spring season? In my understanding mid-April and later is spring. So the sentence does not make sense to me.**
Thank you for the valuable comment – we have corrected the sentence so that it makes more sense: Line 342-343: *"Weekly averages showed a clear decrease from mid-April relative to concentrations in February and March (Figure 1)."*

37. **l. 323: Would the pollution from the military not result in a separate PMF factor? Or is the HOA that is long-range transported so similar to the fresh HOA?**
We estimated the contribution from the local military camp to be 1% of OA as discussed in section 3.2, lines 424-430: *"It is not trivial to distinguish local events and in this case, the possible local contamination was investigated by comparing high HOA peaks (> 0.45 µg m$^{-3}$) with size distribution measurements from the SMPS (Lange et al., 2018). Periods which were attributed to local contamination accounted for less than 1% of OA concentration. Therefore, essentially the entire HOA concentration is assigned to long-range transportation, possibly sources with different ratios of HOA and rBC which would explain the moderate correlation between HOA and rBC."* In general, PMF is not the optimal tool for handling factors of abundances smaller than a few percent.

38. **l. 331: Is this also true for winter? Are there birds all year around? l. 335: Add a reference for the longer lifetime.**
We thank the Referee for the comment and have correct the sentence: Line 353-356: *"In contrast, ammonia (NH$_3$) which is the precursor of NH$_4^+$, derives largely in winter and spring from long-range transport of emissions from biomass burning and agriculture (Fisher et al., 2011), whereas in summertime NH$_3$ emission from seabird-colonies can play a significant role (Croft et al., 2016)."*

We have also added a reference for the longer lifetime of particle bound ammonium: Line 359-360: *"Particle bound NH$_4^+$ has a much longer lifetime than NH$_3$ (Baek and Aneja, 2004) and therefore it is transported as NH$_4^+$ even to the high Arctic."*

39. **l. 336: Please correct Cl to Cl- throughout the manuscript.**
This has been corrected throughout the manuscript (e.g. line 361).

40. **l. 339: should be chloride and not chlorine**
This has now been corrected and more information has been added: Line 363-366: *"However, the SP-AMS does not typically measure refractory chloride at normal vaporizer temperatures, such as NaCl (Canagaratna et al., 2007). Although, Ovadnevaite et al. (2012) has demonstrated how the AMS could be calibrated to measure NaCl in high-time resolution."*

41. **l. 361: Is this true that the sources are the same for the entire Arctic, for all seasons or the Haze period where you have long lifetimes and hence rather well mixed conditions?**
This is a good point and the sentence describes the fact that similar correlation slopes have been observed at different Arctic sites, which suggest similar source regions and not necessarily same sources. We have modified the sentence to make it clear that this is other studies suggestions rather than certain facts: Line 387-389: *"Furthermore, comparable correlation slopes were found for the different Arctic locations, which suggest that source regions of BC and SO$_4^{2-}$ could be similar throughout the Arctic."*

The sources of particles are not the same in the entire Arctic. This has been demonstrated several times latest in; *Dall'Osto, M. Beddows, D.C.S. Tunved, P. Harrison, R. M. Lupi, A. Vitale, V. Becagli, S.*

*Traversi, R. Park, K.T. Yoon, Y.J. Massling, A. Skov, H. Stroam, J. and Krejci, R. (2019). Apportioning aerosol natural and anthropogenic sources thorough simultaneous aerosol size distributions and chemical composition in the European high Arctic. ACP 19, 7377–7395, 2019.* [https://doi.org/10.5194/acp-2018-447](https://doi.org/10.5194/acp-2018-447)*.*

**42. l. 364 – 366: To me it doesn't make sense to include local contamination periods for a general conclusion on rBC and SO4 correlation. I suggest removing the local influence first and then redoing the correlation analysis.**

This is a valid suggestion; however, we have tried to remove the local influences before doing the correlation analysis and it doesn't change the result. We prefer not to leave out any data when correlating rBC and $SO_4^{2-}$ since doing so could result in false security thinking local pollution is completely left out of the correlation. This cannot be guaranteed since we with the current dataset cannot be sure if have "caught" all the local pollution.

**43. l. 407: "AOA is abundant during February to mid-April..." this is redundant. The sentences before that say the same.**

We thank the Referee for this comment and have deleted the repetition (line 443).

**44. l. 421: I cannot follow the argument. What is the contribution quantitatively and what would be expected from the literature? Is the literature appropriate for a comparison?**

The marker ions for BBOA are not specific. SOA also contributes. The argument is that the measured concentration of $C_2H_4O_2^+$ was similar to the amount which is expected from SOA. The argument is rephrased to: Line 455-457: *"However, SOA also contributed to the abundance of $C_2H_4O_2^+$ (Aiken et al., 2008; Aiken et al., 2009; Cubison et al., 2011; Lee et al., 2010; Saarnio et al., 2013). Quantitatively, the expected abundance of C2H4O2+ from SOA did not exceed the measured concentration in this study."*

**45. l. 433: Please be more specific in how far it resembles the Mace Head spectrum.**

We thank the Referee for the comment and the paragraph has been extended: Line 468-475: *"The MOA spectrum resembles a marine organic plume previously published from Mace Head, in the North East Atlantic Ocean showing evidence of both primary and secondary organic aerosols of marine origin (Ovadnevaite et al., 2011). Most abundant peaks in this spectrum were oxygenated fragments at m/z 28 and 44. Also prominent were m/z 27, 39 and 41 from the CH family, and m/z 43 and 55 from the CHO family, which are also found in the MOA spectrum. The two spectra differ in terms of abundances of CH-like organic matter, but they are different from the marine organic aerosol factor published during the ASCOS expedition in the Central Arctic Ocean (Chang et al., 2011), which shows a closer resemblance with the mass spectrum of pure MSA, i.e. dominating peaks at m/z 15, 48, 64 and 79."*

**46. l. 443: How does the MOA factor resemble HR-AMS spectra from the Southern Ocean? doi:10.5194/acp-13-8669-2013 Can the authors discuss whether the MOA factor is more universal, i.e. VRS, Mace Head, other oceans?**

We find MOA to resemble the marine bloom at Mace Head in the north east Atlantic Ocean, which is interesting since the two marine environments are located not too far away from each other. Comparisons with Southern Oceans may be somewhat out of scope.

**47. l. 456: What is the lowest concentration of OA?**

The lowest concentration of OA during May where MOA is dominant is 0.01 µg/m$^3$, which is now added to the sentence: Line 493-494: *"At the same time, we observe the lowest concentration of OA (0.01 µg/m3) consisting of 75% MOA (Figure 4)."*

**48. l. 456: What does "this" refer to? The concentration of OA or the 75 % MOA in the OA?**

We have corrected the sentence so that is clearer: Line 494-497: *"This is significantly higher than observed at Alert by Narukawa et al. (2008) where marine organic matter contributed 45% to aerosol total carbon in late spring (26 April – 6 May 2000). However, direct comparison is difficult due to different methods and time periods (Narukawa et al., 2008)."*

**49. l. 475ff: This sentence is confusing. I do not understand the main message.**
We thank the Referee for this valuable comment and have therefore changed the sentence to accommodate the suggestions: Line 517-522: *"MOA constitutes 22% of OA on average during our measurement period ranging from 2-3% of OA in February and March to 24% and 74% of OA in April and May, respectively (Figure 2b and 4). Thus, MOA is by far the most abundant OA from end of April and onwards. MOA dominates the OA mass after polar sunrise and persists during polar daytime so the aerosol's optical impact might be substantial. At the same time, MOA dominates when the overall $PM_1$ concentration is very low, particle numbers are low and hence CCN concentrations can be low."*

**50. l. 480: "oxidation products of DMS and other VOCs" These are also secondary. The argument does not make sense like this.**
We thank the Referee for finding this mistake and we have now corrected the sentence: Line 524- 527: *"MOA may contain oxidation products of DMS and other VOCs from oceanic origin, as well as a variety of primary components including sacharides such as mannitol in addition to insoluble gels (Croft et al., 2018; Leck and Bigg, 2005; Orellana et al., 2011; Fu et al., 2013; Ovadnevaite et al., 2011)."*

**51. l. 481: "And primary components including colloidal gels..." As far as I read the sentence MOA is the specific factor found by the authors using the HR-AMS. So the question is whether the primary compounds like gels would actually be seen in the MOA factor? To my knowledge they evaporate at temperatures higher than 600 C. This means that generally marine organic aerosol can contain these compounds, but the MOA factor likely doesn't due to instrumental limitations.**
The sentence has been rewritten: Line 524-527: *"MOA may contain oxidation products of DMS and other VOCs from oceanic origin, as well as a variety of primary components including sacharides such as mannitol in addition to insoluble gels (Croft et al., 2018; Leck and Bigg, 2005; Orellana et al., 2011; Fu et al., 2013; Ovadnevaite et al., 2011)."*

**52. l. 487: enhancement through the lensing effect?**
We have modified and moved this part to the introduction where it fits better: Line 114-117: *"Biogenic marine aerosols can scatter solar radiation, which will result in a negative radiative forcing. Biogenic marine aerosols can also coat soot particles, which may be transported from wild fires (AMAP, 2015), which could impact the CCN activity and absorption by the soot particles (Lange et al., 2018)."*

**53. l. 494f: 75+3+12+12is>100%.**
After checking the reference (Udisti et al., 2016) we have changed the sentence to: Line 536-538: *"This is in accordance with previous findings from VRS, Alert (Norman et al., 1999) and Svalbard (Udisti et al., 2016) where $SO_4^{2-}$ has been apportioned to be 65% and 75% anthropogenic, respectively."*

**54. l. 503 What does "reduced" mean? The least amount of oxygen?**
Yes, and we have hence corrected the sentence: Line 545: *"HOA, being the least oxidized factor, made up 12% of OA..."*

**55. Figure 2: I suggest to either make the axis logarithmic or put them off at 0.05 (with indicating the true extent of the big peaks) to make the pattern visible. The AOA and MOA spectra are not informative like they are now because on cannot see anything.**
We have split the y-axis for AOA and MOA in order to make the pattern more visible as requested (line 1049).

**56. Figure 3: I suggest to move the rBC trace up. It's not visible like this and hence not useful.**
This is a valid point and we have changed the y-axis for rBC from [0;3] to [-1;3], which makes rBC visible (line 1053).

**57. Figures S3: the figures have very low resolution. Figure S2: The y-axis could start at 0.5.**

We have improved the resolution in Figure S3 (line S56) and changed the y-axis on Figure S2 so that it starts at 0.5 instead of 0 (Line S53).

**58. L.437-440: This discussion does not reveal to me in how far the spectrum at VRS contains MSA tracers. Please clarify.**

We thank the Referee for the comment and have now added that we observe MSA in our MOA factor: Line 474-475: *"… which shows a closer resemblance with the mass spectrum of pure MSA, i.e. dominating peaks at m/z 15, 48, 64 and 79."*

**Please also note the supplement to this comment: https://www.atmos-chem-phys-discuss.net/acp-2019-130/acp-2019-130-RC3-supplement.pdf**

We have gone through the additional comments provided in supplement from the Referee. We have incorporated the suggested changes, which are summarized shortly below. One comment in this supplement we found too substantial so we have added it under specific comments as no. 59 (see above).

1. We have changed *"To address this, we report 93 days of Soot Particle Aerosol Mass Spectrometer (SP-AMS) data collected in the high Arctic. The period spans from February 20$^{th}$ until May 23$^{rd}$ 2015 at Villum Research Station (VRS) in Northern Greenland (81°36' N)"* to *"To address this, we report 93 days of Soot Particle Aerosol Mass Spectrometer (SP-AMS) data collected from February 20$^{th}$ until May 23$^{rd}$ 2015 at Villum Research Station (VRS) in Northern Greenland (81°36' N)"* (line 21-23).
2. We have deleted *"Important differences are observed among the factors, including the"* and *"= the marine related factor"* (line 39).
3. We have deleted *"the"* (line 62).
4. We have deleted *"exchange"* (line 68).
5. We have deleted *", which amounted"* (line 82).
6. We have modified the sentence from: *"BC is deposited on snow and ice-covered surfaces it changes the albedo, leading to increased absorption of solar radiation and direct heating of the surface"* to *"BC deposited on snow and ice-covered surfaces changes the albedo, leading to increased absorption of solar radiation and direct heating of the surface"* (line 88-89).
7. We have deleted *"formation"* (line 125).
8. We have deleted *"and particle size distribution, respectively"* (line 184).
9. We have changed *"span"* to *"range"* (line 255).
10. We have deleted *"a"* (line 266).
11. We have deleted *"from the SP-AMS"* (line 293).
12. We have deleted *"measured by the SP-AMS"* (line 318).
13. We have deleted "aerosols" (line 321).
14. We have deleted *"dominated by"* (line 322).
15. We have deleted *"in"* (line 339).
16. We have deleted *"ratio"* (line 467).
17. We have changed *"illustrates"* to *"illustrate"* (line 482).
18. We have deleted *", only, above the mountains at the horizon"* (line 484).
19. We have decided to keep *"Northern Hemisphere"* even though the Referee suggests to delete it (line 489).
20. We have decided to keep *"Northern Hemisphere"* even though the Referee suggests to delete it (line 498).
21. We have changed *"sea-ice"* to *"sea ice"* (line 500).

[revised manuscript text omitted]